# A cluster-randomized trial of water, sanitation, handwashing and nutritional interventions on stress and epigenetic programming

A regulated stress response is essential for healthy child growth and development trajectories. We conducted a cluster-randomized trial in rural Bangladesh (funded by the Bill & Melinda Gates Foundation, ClinicalTrials.gov NCT01590095) to assess the effects of an integrated nutritional, water, sanitation, and handwashing intervention on child health. We previously reported on the primary outcomes of the trial, linear growth and caregiver-reported diarrhea. Here, we assessed additional prespecified outcomes: physiological stress response, oxidative stress, and DNA methylation ($N = 759$, ages 1–2 years). Eight neighboring pregnant women were grouped into a study cluster. Eight geographically adjacent clusters were block-randomized into the control or the combined nutrition, water, sanitation, and handwashing (N + WSH) intervention group (receiving nutritional counseling and lipid-based nutrient supplements, chlorinated drinking water, upgraded sanitation, and handwashing with soap). Participants and data collectors were not masked, but analyses were masked. There were 358 children (68 clusters) in the control group and 401 children (63 clusters) in the intervention group. We measured four F2-isoprostanes isomers (iPF(2α)-III; 2,3-dinor-iPF(2α)-III; iPF(2α)-VI; 8,12-iso-iPF(2α)-VI), salivary alpha-amylase and cortisol, and methylation of the glucocorticoid receptor (*NR3C1*) exon 1F promoter including the NGFI-A binding site. Compared with control, the N + WSH group had lower concentrations of F2-isoprostanes isomers (differences ranging from −0.16 to −0.19 log ng/mg of creatinine, $P < 0.01$), elevated post-stressor cortisol (0.24 log μg/dl; $P < 0.01$), higher cortisol residualized gain scores (0.06 μg/dl; $P = 0.023$), and decreased methylation of the NGFI-A binding site (−0.04; $P = 0.037$). The N + WSH intervention enhanced adaptive responses of the physiological stress system in early childhood.

Children living in low-income settings often experience recurrent infections and undernutrition due to inadequate water, sanitation, and hygiene infrastructure and food insecurity that may have lasting impacts on their stress response system critical for healthy growth and development. Although several studies have evaluated the effects of psychosocial interventions on the physiological stress system[1], a major gap is the lack of experimental studies assessing the effects of physical health interventions on the hypothalamic-pituitary-adrenocortical

✉e-mail: audrielin@ucsc.edu

(HPA) and sympathetic adrenomedullary (SAM) systems and epigenetic programming in early childhood.

Chronic stress, in the form of undernutrition, infection, and psychosocial adversity, may cause irreversible harm if it occurs during the early years of life (under age 2 years), a period of rapid growth and development[2]. During this period of heightened plasticity, the neuroendocrine-immune network develops and adapts in response to exposure to environmental stimuli[3]. Stressful stimuli shape the set point, reactivity, and regulation of the two primary neuroendocrine axes—the SAM and the HPA systems—and these axes, in turn, regulate the immune system[3]. Activation of the SAM system leads to increased blood pressure and heart rate and changes in the levels of salivary alpha-amylase, a biomarker of the SAM system[4,5].

The HPA axis modulates the SAM system through the production of glucocorticoids. Cortisol, a key glucocorticoid, regulates the immune system, growth factors, and neurodevelopment[6–13]. Cortisol production follows a circadian rhythm that is developed during the first year of life[14]. Chronic stress disrupts the tight regulation of this circadian rhythm[15]. An HPA or SAM challenge, such as an acute physical stressor (e.g., vaccination), is typically used to induce and measure cortisol or salivary alpha-amylase reactivity in children[16]. This artificial induction of reactivity reflects the magnitude of an individual's HPA or SAM response to a stressor in a naturalistic setting[17,18]. Exposure to chronic stress may alter a child's cortisol response to an acute physical stressor, which could indicate HPA axis dysregulation. Glucocorticoids also regulate genes involved in oxidative stress pathways[19,20]. Oxidative stress results from an imbalance between generation of reactive oxygen species and elimination of them through the antioxidant defense system[21]. F2-isoprostanes reflect systemic oxidative damage[22,23] and are associated with infections and neurological damage[24,25].

Cortisol binds to the glucocorticoid receptor, encoded by the *NR3C1* gene[6]. Studies suggest associations between stress and epigenetic modulation of the *NR3C1* gene[6]. During early childhood, the epigenome undergoes dramatic changes. Environmental factors significantly affect these epigenetic processes, which involve the regulation of gene expression through methylation and chromatin modification. Recent studies are beginning to elucidate the degree to which environmental factors during infancy affect developmental programming at the DNA level to determine health outcomes in adulthood. Within the *NR3C1* gene, increased methylation of CpG sites of a noncanonical nerve growth factor-inducible protein A (NGFI-A) binding site downregulates the expression of the *NR3C1* gene[26]. Differential methylation of *NR3C1* or the NGFI-A binding site is associated with childhood maltreatment[27].

Previously, the WASH Benefits trial reported that children receiving a combined nutrition, water, sanitation, and handwashing intervention experienced better growth (primary outcome), reduced diarrhea prevalence (primary outcome), and improved neurodevelopment (secondary outcome) compared to children in the control group[28,29]. Here, we evaluated the effects of the intervention on additional prespecified outcomes: physiological stress response, oxidative stress levels, and DNA methylation of the *NR3C1* gene among young children.

## Results
### Enrollment
The overall trial assessed 13279 pregnant women for eligibility. 5551 women were enrolled between 31 May 2012 and 7 July 2013. A total of 90 blocks were defined, and each block was made up of 8 clusters randomly allocated to one of the intervention or control groups (720 clusters randomized) (Fig. 1). The target enrollment for the stress substudy was 996 children after one year of intervention and 1021 children after two years of intervention (Fig. 1). The substudy included 70 blocks (135 clusters). Stress outcomes were assessed in 688 children (51% female) at age 14.3 (IQR, 12.7–15.6) months, and 759 children (52%

female) at age 28.2 (IQR, 27.0–29.6) months (Fig. 1). At enrollment, household characteristics were similar across intervention and control arms (Table 1) and comparable to the overall trial (Supplementary Table 1).

### Oxidative stress
Compared to children in the control group, after one year of combined N + WSH intervention (median age 14 months), children in the intervention arm exhibited lower levels of all four F2-isoprostanes isomers measured: IPF(2α)-III (−0.16 log ng/mg of creatinine; CI −0.27 to −0.06, $P < 0.01$), 2,3-dinor-iPF(2α)-III (−0.16 log ng/mg of creatinine; CI −0.23 to −0.09, $P < 0.001$), iPF(2α)-VI (−0.17 log ng/mg of creatinine; CI −0.25 to −0.1, $P < 0.001$), and 8,12-iso-iPF(2α)-VI (−0.19 log ng/mg creatinine; CI −0.29 to −0.1, $P < 0.001$) (Table 2; Supplementary Fig. 1; Supplementary Table 2).

### HPA and SAM axes
For the cortisol and salivary alpha-amylase measurements, the acute stressor was a venipuncture and physical separation of the child from the caregiver. In terms of cortisol reactivity, 69% of the children were classified as responders to the acute stressor (exhibited an increase in pre- to post-stressor cortisol levels), 13% of the children were classified as non-responders to the acute stressor (exhibited a decrease in pre- to post-stressor cortisol levels), and 18% experienced no change in cortisol levels (Supplementary Fig. 2). After two years of intervention (median age 28 months), children in the combined N + WSH intervention group had elevated post-stressor salivary cortisol levels (0.24 log μg/dl; CI 0.07 to 0.4, $P < 0.01$), higher cortisol slope scores (0.002 μg/dl/min; CI 0 to 0.003, $P = 0.035$), and higher residualized gain scores for cortisol (0.06 μg/dl; CI 0.01 to 0.12, $P = 0.023$; Table 3; Supplementary Fig. 1; Supplementary Table 3). There was no difference in the overall methylation levels of the *NR3C1* between the control and the combined N + WSH intervention groups. Logit-transformed methylation of the NGFI-A transcription factor binding site was lower in the combined N + WSH intervention group compared to the control group (−0.04, CI −0.08 to 0, $P = 0.037$; Table 3; Supplementary Fig. 1; Supplementary Table 3). Unadjusted, adjusted, and inverse probability of censoring weighted (IPCW) analyses produced similar estimates (Supplementary Tables 2 and 3), indicating balance in measured confounders across arms and no differential loss to follow-up.

### Subgroup analyses
A prespecified subgroup analysis revealed that sex was not an effect modifier for F2-isoprostanes at year one (Supplementary Table 4). After two years, there was some evidence of effect measure modification with child sex: among males, the combined N + WSH intervention group had a lower resting heart rate compared to the control group (−3.53 bpm, CI −6.62 to −0.44; $P = 0.025$), and there was no effect among females (sex by treatment interaction $P = 0.027$; Supplementary Table 5). Among females, the combined N + WSH intervention group had higher pre-stressor cortisol levels compared to the control group (0.16 log μg/dl, CI 0.01 to 0.32; $P = 0.038$), and there was no intervention effect on pre-stressor cortisol levels among males (sex by treatment interaction $P = 0.035$).

## Discussion
The trial found that a combined nutrition, water, sanitation, and handwashing intervention reduced oxidative stress, enhanced HPA axis functioning, and reduced methylation levels of the NGFI-A binding site in the *NR3C1* exon 1F promoter in young children. The magnitude of the effects of this environmental and nutritional intervention on cortisol production is within the range of intervention effects of psychosocial interventions reported in early childhood[30]. The N + WSH intervention effects on F2-isoprostanes are also comparable to the effects of dietary interventions in adult populations[31].

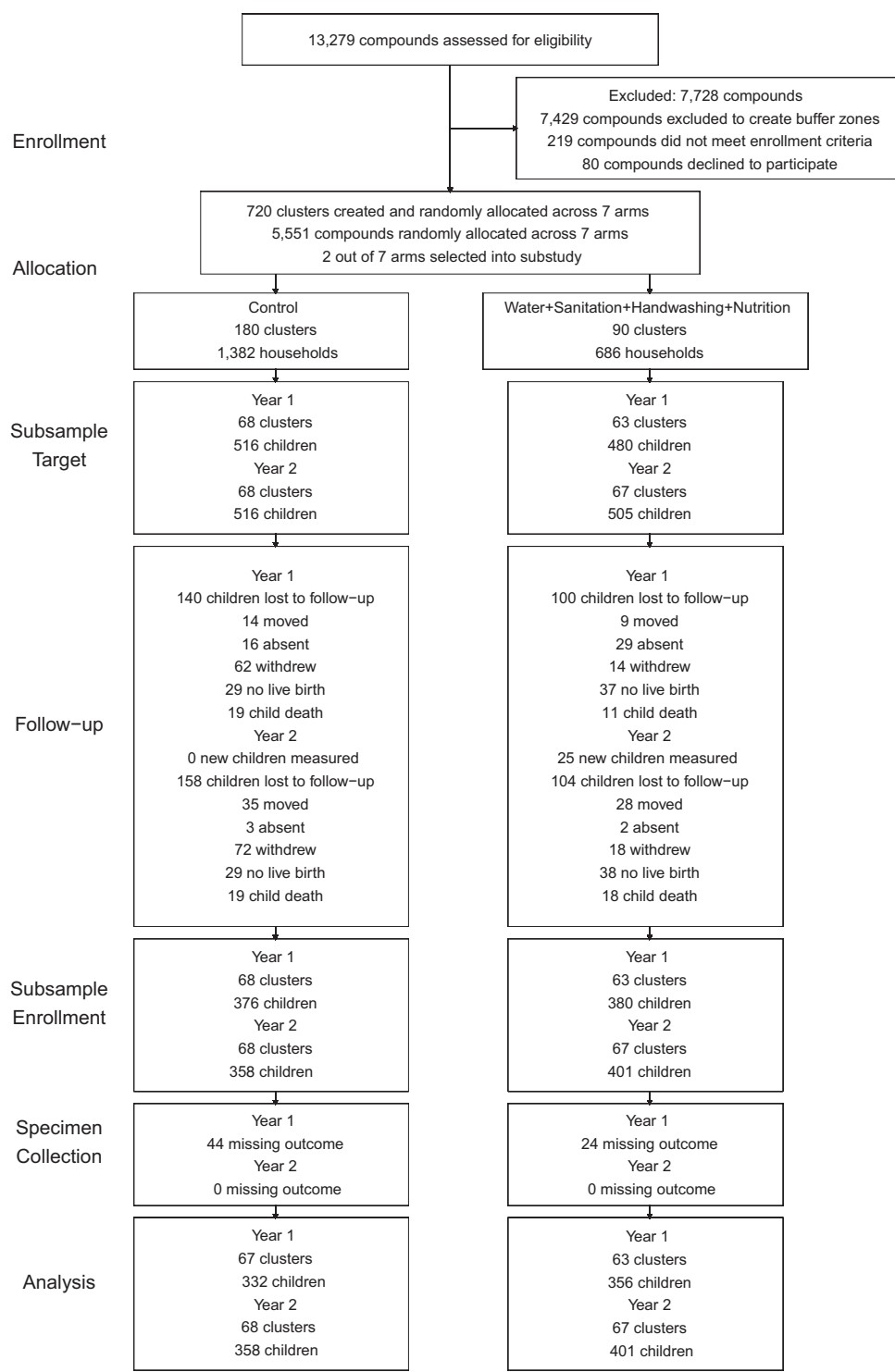

**Fig. 1 | Participant enrollment for the WASH Benefits stress response and DNA methylation study population.** Source data are provided as a Source Data file.

Oxidative stress, the accumulation of unstable free radicals that damage DNA and cellular structures, has been implicated in the pathophysiology of several pediatric disorders including asthma, protein-energy malnutrition, and diarrheal diseases[32]. Augmented oxidative stress in children with severe forms of malnutrition, including kwashiorkor and marasmus, may be the consequence of increased production of reactive oxygen species or impaired anti-oxidant defenses[33,34]. Micronutrients serve key roles in the body's antioxidant defense system, either directly as antioxidants (e.g., vitamins C, A, and E) or indirectly as co-factors of antioxidant enzymes (e.g., manganese, copper, or zinc)[35]. The combined N + WSH intervention reduced diarrhea, anemia, iron deficiency, and ulti-mately improved child growth[28,36]. The lipid nutrient supplement containing ≥100% of the recommended daily allowance for 12 vita-mins, including antioxidants such as vitamins C, A, and E and 9 minerals including manganese, copper, zinc, and selenium (full list available in Stewart et al.[36]) may have strengthened the antioxidant defenses of children in the intervention group compared to children in the control group. With enhanced antioxidant defenses, the bodies of children who received the intervention may have more efficiently

**Table 1 | Enrollment characteristics within the Control households and the N + WSH intervention households among children measured 1 year and among children measured 2 years post-intervention**

| No. of compounds: | Children measured at Year 1 | | Children measured at Year 2 | |
|---|---|---|---|---|
| | Control (N = 332) | N + WSH Intervention (N = 356) | Control (N = 358) | N + WSH Intervention (N = 401) |
| Maternal | | | | |
| Age(years) | 23.4 (4.9) | 24.4 (5.5) | 23.4 (4.9) | 24.2 (5.5) |
| Years of education | 6.8 (3.3) | 5.8 (3.5) | 6.7 (3.2) | 5.8 (3.4) |
| Paternal | | | | |
| Years of education | 5.6 (3.9) | 4.7 (4) | 5.5 (3.9) | 4.8 (4) |
| Works in agriculture | 80 (24%) | 101 (28%) | 86 (24%) | 114 (28%) |
| Household | | | | |
| Number of people | 4.9 (2.5) | 4.9 (2.1) | 4.8 (2.4) | 4.9 (2.2) |
| Has electricity | 200 (60%) | 214 (60%) | 218 (61%) | 248 (62%) |
| Has a cement floor | 59 (18%) | 43 (12%) | 59 (16%) | 51 (13%) |
| Acres of agricultural land owned | 0.2 (0.3) | 0.1 (0.1) | 0.2 (0.2) | 0.1 (0.1) |
| Drinking Water | | | | |
| Shallow tubewell primary water source | 236 (71%) | 249 (70%) | 263 (73%) | 291 (73%) |
| Stored water observed at home | 168 (51%) | 185 (52%) | 176 (49%) | 205 (51%) |
| Reported treating water yesterday | 1 (0%) | 1 (0%) | 1 (0%) | 1 (0%) |
| Distance (mins) to primary water source | 0.8 (1.6) | 0.8 (2.3) | 0.7 (1.5) | 0.7 (2.2) |
| Sanitation | | | | |
| Reported daily open defecation | | | | |
| Adult men | 12 (4%) | 31 (9%) | 15 (4%) | 34 (9%) |
| Adult women | 8 (2%) | 16 (5%) | 10 (3%) | 19 (5%) |
| Children: 8 to <15 years | 6 (4%) | 18 (11%) | 4 (3%) | 19 (10%) |
| Children: 3 to <8 years | 47 (29%) | 68 (33%) | 54 (31%) | 80 (36%) |
| Children: 0 to <3 years[a] | 55 (71%) | 69 (88%) | 62 (75%) | 73 (88%) |
| Latrine | | | | |
| Owned[b] | 208 (63%) | 191 (54%) | 209 (58%) | 212 (53%) |
| Concrete Slab | 315 (97%) | 317 (93%) | 339 (96%) | 356 (93%) |
| Functional water seal | 118 (39%) | 89 (32%) | 123 (38%) | 97 (30%) |
| Visible stool on slab or floor | 149 (46%) | 177 (53%) | 162 (46%) | 200 (54%) |
| Owned a child potty | 27 (8%) | 16 (4%) | 25 (7%) | 20 (5%) |
| Human feces observed in the | | | | |
| House | 23 (7%) | 30 (8%) | 20 (6%) | 34 (9%) |
| Child's play area | 4 (1%) | 6 (2%) | 4 (1%) | 6 (2%) |
| Handwashing location | | | | |
| Within six steps of latrine | | | | |
| Has water | 57 (19%) | 45 (14%) | 70 (21%) | 52 (14%) |
| Has soap | 30 (10%) | 20 (6%) | 36 (11%) | 25 (7%) |
| Within six steps of kitchen | | | | |
| Has water | 32 (10%) | 33 (10%) | 41 (12%) | 40 (11%) |
| Has soap | 12 (4%) | 9 (3%) | 17 (5%) | 14 (4%) |
| Nutrition | | | | |
| Household is food secure[c] | 246 (74%) | 251 (71%) | 264 (74%) | 286 (71%) |

Data are n (%) or mean (SD). Percentages were estimated from slightly smaller denominators than those shown at the top of the table for the following variables due to missing values: mother's age, father's education, father works in agriculture, acres of land owned, open defecation, latrine has a concrete slab, latrine has a functional water seal, visible stool on latrine slab or floor, ownership of child potty, observed feces in the house or child's play area, handwashing variables.
[a]Open defecation does not include diaper disposal of feces.
[b]Households who do not own a latrine typically share a latrine with extended family members who live in the same compound.
[c]Assessed by the Household Food Insecurity Access Scale.
Source data are provided as a Source Data file.

scavenged free radicals, prevented their formation, and disrupted free-radical reactions.

As part of the innate immune response against pathogen invasion, phagocytic cells release reactive oxygen and nitrogen species that target proteins, DNA, and lipids[37]. Innate immunity is tightly regulated because reactive oxygen and nitrogen species target pathogens and host cells alike. After one year, children in the intervention arm experienced less diarrhea and less enteric viral infections[28,38]. It is plausible that the water, sanitation, and handwashing interventions interrupted pathogen transmission, leading to lower non-specific

**Table 2 | Effect of nutrition, water, sanitation, and handwashing intervention on oxidative stress measurements on Bangladeshi children at age 14 months**

| Urinary F2-isoprostanes | N | Absolute Mean | Mean | SD | Unadjusted difference: Intervention v. Control | | Age- and sex-adjusted difference: Intervention v. Control | | Fully adjusted difference: Intervention v. Control[a] | |
|---|---|---|---|---|---|---|---|---|---|---|
| | | | | | 95% CI | P-value | 95% CI | P-value | 95% CI | P-value |
| Ln iPF(2α)-III (ng/mg creatinine) | | | | | | | | | | |
| Control | 332 | 0.83 | −0.33 | 0.54 | | | | | | |
| Nutrition + WSH | 356 | 0.7 | −0.5 | 0.56 | −0.16 (−0.27, −0.06) | 0.002 | −0.17 (−0.28, −0.06) | 0.002 | −0.14 (−0.22, −0.05) | 0.001 |
| Ln 2,3-dinor-iPF(2α)-III (ng/mg creatinine) | | | | | | | | | | |
| Control | 332 | 6.66 | 1.84 | 0.33 | | | | | | |
| Nutrition + WSH | 356 | 5.64 | 1.68 | 0.32 | −0.16 (−0.23, −0.09) | <0.001 | −0.19 (−0.26, −0.13) | <0.001 | −0.18 (−0.24, −0.12) | <0.001 |
| Ln iPF(2α)-VI (ng/mg creatinine) | | | | | | | | | | |
| Control | 332 | 16.36 | 2.68 | 0.46 | | | | | | |
| Nutrition + WSH | 356 | 13.5 | 2.5 | 0.44 | −0.17 (−0.25, −0.1) | <0.001 | −0.2 (−0.27, −0.13) | <0.001 | −0.19 (−0.25, −0.12) | <0.001 |
| Ln 8,12-iso-iPF(2α)-VI (ng/mg creatinine) | | | | | | | | | | |
| Control | 332 | 16.05 | 2.63 | 0.55 | | | | | | |
| Nutrition + WSH | 356 | 13.49 | 2.44 | 0.57 | −0.19 (−0.29, −0.1) | <0.001 | −0.21 (−0.31, −0.12) | <0.001 | −0.24 (−0.32, −0.16) | <0.001 |

We used targeted maximum likelihood estimation with influence curve-based standard errors accounting for clustered observations from the trial's geographic block-randomized design.
Confidence intervals were adjusted for clustered observations using robust standard errors. The statistical tests were two-sided, and no adjustments were made for multiple comparisons.
[a]Adjusted for prespecified covariates: child sex, child birth order, mother's age, mother's height, mother's education, number of children <18 years in the household, number of individuals living in the compound, distance in minutes to the primary water source, household food security, household floor materials, household wall materials, household electricity, and household assets (wardrobe, table, chair, clock, khat, chouki, radio, television, refrigerator, bicycle, motorcycle, sewing machine, mobile phone, cattle, goats, and chickens), child age at dates of urine, vitals, and saliva collection, and monsoon season at dates of urine, vitals, and saliva collection.
Source data are provided as a Source Data file.

innate immune activity, less generation of reactive oxygen species, and reductions in downstream lipid peroxidation as measured by F2-isoprostanes. Together, less immune-activated production of reactive oxygen species and reinforced antioxidant defenses may have contributed to the lower levels of oxidative stress observed in children receiving the combined N + WSH interventions compared to children in the control group. These biological mechanisms could underpin the subsequent improvements in child development that we observed at age two years in the intervention group[29].

The effects of the intervention were remarkably consistent across upstream and downstream levels of the HPA axis. Compared to the control group, the combined N + WSH intervention group had hypomethylated NGFI-A transcription factor binding site, which leads to elevated glucocorticoid receptor gene expression[39]. The higher post-stressor cortisol levels and cortisol reactivity exhibited in the intervention group are also indicative of increased activation of the glucocorticoid receptor. The mechanisms by which nutrition modulates the HPA axis have not been fully elucidated but could be mediated through the immune triad[15] or epigenetic programming[40].

In settings with inadequate water, sanitation, and hygiene infrastructure, infections are acquired early and frequently in childhood. Infections affect the transcriptional activation of the glucocorticoid receptor[41]. In the WASH Benefits trial, children in arms receiving the WSH intervention had reduced *Giardia* and hookworm infections and lower acute respiratory illness at age two years compared with children in the control group[42–44]. The hypomethylation of the NGFI-A transcription factor binding site and the robust cortisol response observed in the intervention group may have strengthened resistance to infections, as glucocorticoids are immunomodulatory hormones[45]. Sex assigned at birth was an effect modifier: among females, the intervention group experienced an elevation in pre-stressor cortisol levels. Identifying mediators of these sex-based differences in stress response will guide the future design of targeted early childhood interventions. During early childhood, a period of increased physiological plasticity

and epigenetic programming, the combined nutrition, water, sanitation, and handwashing intervention enhanced children's HPA axis regulatory capabilities, which in turn may have cascading effects on growth and development.

In the parent trial, improved child linear growth was only observed in the arms receiving the nutrition intervention (the nutrition alone arm and the combined N + WSH arm)[28], and child neurodevelopment was improved in all intervention arms[29]. Although the combined N + WSH intervention may have directly impacted the physiological stress system through improved nutritional status and infection reduction, the intervention may also have had indirect impacts through reductions in caregiver depression and enhanced caregiver social support. In the WASH Benefits parent trial, community health promoters visited households in the intervention arm to promote intervention-related behaviors (e.g., treating water, using latrines), but they did not visit households in the control arm. The trial reported lower levels of depressive symptoms among women in the intervention arm compared with the control arm[29]. Maternal depression is associated with blunted cortisol reactivity in infants, mediated by increased family conflict and less responsive parenting[46]. Hence, the intervention-associated reductions in maternal depressive symptoms and increased caregiver support may have led to the higher cortisol reactivity observed among children in the intervention arm compared with the children in the control arm. Forthcoming studies will aim to elucidate the complex interplay between nutrition, infection, psychosocial factors, and the physiological stress response.

This study has limitations. One limitation of the study is that we only analyzed cortisol and salivary alpha-amylase reactivity, which prevented us from characterizing the full cortisol and salivary alpha-amylase awakening response, a common measure of HPA axis and autonomic nervous system functioning. Here, we report the effects of the combined intervention on stress response, because we did not also analyze samples from children who received only the nutrition and

**Table 3 | Effect of water, sanitation, handwashing, and nutrition intervention on stress response and DNA methylation measurements on Bangladeshi children at age 28 months**

| Outcome | N | Absolute Mean | Mean | SD | Unadjusted difference: Intervention v. Control | | Age- and sex-adjusted difference: Intervention v. Control | | Full adjusted difference: Intervention v. Control[a] | |
|---|---|---|---|---|---|---|---|---|---|---|
| | | | | | 95% CI | P-value | 95% CI | P-value | 95% CI | P-value |
| Ln pre-stressor salivary alpha-amylase (U/ml) | | | | | | | | | | |
| Control | 354 | 74.9 | 4.01 | 0.81 | | | | | | |
| Nutrition + WSH | 394 | 75.92 | 3.98 | 0.89 | −0.03 (−0.19, 0.13) | 0.731 | −0.02 (−0.18, 0.14) | 0.785 | −0.02 (−0.17, 0.14) | 0.834 |
| Ln post-stressor salivary alpha-amylase (U/ml) | | | | | | | | | | |
| Control | 339 | 124.06 | 4.47 | 0.9 | | | | | | |
| Nutrition + WSH | 375 | 122.53 | 4.41 | 0.99 | −0.06 (−0.26, 0.13) | 0.530 | −0.06 (−0.24, 0.13) | 0.550 | −0.01 (−0.2, 0.18) | 0.928 |
| Slope between pre- and post-stressor salivary alpha-amylase (U/ml/min) | | | | | | | | | | |
| Control | 335 | 2.99 | 2.99 | 6.43 | | | | | | |
| Nutrition + WSH | 367 | 2.73 | 2.73 | 5.29 | −0.22 (−1.07, 0.64) | 0.622 | −0.2 (−1.02, 0.62) | 0.635 | 0.09 (−0.72, 0.91) | 0.821 |
| Residualized gain score for alpha-amylase (U/ml) | | | | | | | | | | |
| Control | 335 | 0.77 | 0.77 | 108.21 | | | | | | |
| Nutrition + WSH | 368 | −0.7 | −0.7 | 89.31 | −1.17 (−15.16, 12.82) | 0.870 | −0.19 (−13.82, 13.45) | 0.979 | 4.97 (−8.79, 18.72) | 0.479 |
| Ln pre-stressor salivary cortisol (µg/dl) | | | | | | | | | | |
| Control | 357 | 0.17 | −2.08 | 0.69 | | | | | | |
| Nutrition + WSH | 396 | 0.18 | −2.03 | 0.73 | 0.05 (−0.08, 0.18) | 0.458 | 0.04 (−0.09, 0.17) | 0.553 | 0.04 (−0.08, 0.17) | 0.487 |
| Ln post-stressor salivary cortisol (µg/dl) | | | | | | | | | | |
| Control | 312 | 0.34 | −1.49 | 0.96 | | | | | | |
| Nutrition + WSH | 385 | 0.42 | −1.26 | 0.95 | 0.24 (0.07, 0.4) | 0.005 | 0.21 (0.05, 0.37) | 0.010 | 0.21 (0.06, 0.36) | 0.007 |
| Slope between pre- and post-stressor cortisol (µg/dl/min) | | | | | | | | | | |
| Control | 311 | 0.01 | 0.01 | 0.01 | | | | | | |
| Nutrition + WSH | 380 | 0.01 | 0.01 | 0.01 | 0.002 (0, 0.003) | 0.035 | 0.002 (0, 0.003) | 0.053 | 0.002 (0, 0.003) | 0.031 |
| Residualized gain score for cortisol (µg/dl) | | | | | | | | | | |
| Control | 311 | −0.03 | −0.03 | 0.27 | | | | | | |
| Nutrition + WSH | 380 | 0.03 | 0.03 | 0.32 | 0.06 (0.01, 0.12) | 0.023 | 0.06 (0, 0.11) | 0.033 | 0.06 (0.01, 0.11) | 0.018 |
| Mean arterial pressure (mmHg) | | | | | | | | | | |
| Control | 353 | 65.18 | 65.18 | 6.14 | | | | | | |
| Nutrition + WSH | 399 | 65.5 | 65.5 | 6.78 | 0.33 (−1, 1.66) | 0.625 | 0.32 (−1.06, 1.7) | 0.649 | 0.31 (−0.84, 1.46) | 0.596 |
| Resting heart rate (bpm) | | | | | | | | | | |
| Control | 358 | 109.49 | 109.49 | 14.43 | | | | | | |
| Nutrition + WSH | 398 | 108.12 | 108.12 | 17.12 | −1.35 (−4.08, 1.39) | 0.334 | −1.43 (−4.14, 1.27) | 0.299 | −1.65 (−4.26, 0.97) | 0.218 |
| Logit-transformed *NR3C1* exon 1F promoter methylation | | | | | | | | | | |
| Control | 346 | 0.39 | −3.53 | 0.09 | | | | | | |
| Nutrition + WSH | 396 | 0.38 | −3.53 | 0.1 | −0.001 (−0.02, 0.018) | 0.917 | −0.001 (−0.021, 0.019) | 0.895 | −0.003 (−0.02, 0.014) | 0.720 |
| Logit-transformed NGFI-A transcription factor binding site methylation | | | | | | | | | | |
| Control | 336 | 0.9 | −3.39 | 0.25 | | | | | | |
| Nutrition + WSH | 386 | 0.77 | −3.43 | 0.26 | −0.04 (−0.08, 0) | 0.037 | −0.04 (−0.08, 0) | 0.080 | −0.04 (−0.08, 0) | 0.037 |

We used targeted maximum likelihood estimation with influence curve-based standard errors accounting for clustered observations from the trial's geographic block-randomized design. Confidence intervals were adjusted for clustered observations using robust standard errors. The statistical tests were two-sided, and no adjustments were made for multiple comparisons.
[a]Adjusted for prespecified covariates: child sex, child birth order, mother's age, mother's height, mother's education, number of children <18 years in the household, number of individuals living in the compound, distance in minutes to the primary water source, household food security, household floor materials, household wall materials, household electricity, and household assets (wardrobe, table, chair, clock, khat, chouki, radio, television, refrigerator, bicycle, motorcycle, sewing machine, mobile phone, cattle, goats, and chickens), child age at dates of urine, vitals, and saliva collection, and monsoon season at dates of urine, vitals, and saliva collection. For salivary alpha-amylase and cortisol analyses only, time of sampling was included as a covariate.
Source data are provided as a Source Data file.

only the WSH intervention; thus, we cannot determine whether the effects were primarily driven by the nutrition intervention, the WSH intervention, or the combination of both. There was also loss-to-follow-up among children targeted for enrollment in this study and between measurement rounds (Fig. 1). Though loss-to-follow-up may have reduced the study's power, randomization was maintained, as evidenced by the balance between household characteristics across intervention and control arms (both between children with outcome data (Table 1) and between those lost to follow-up (Supplementary Table 1)), suggesting that selection bias from differential loss-to-follow-up was unlikely. Additionally, an inverse probability of censoring-weighted analysis, which re-weights the measured outcomes so the observed population reflects the characteristics of the full study population[47,48], produced similar estimates (Supplementary

Tables 2 and 3), which further suggests that differential loss-to-follow-up likely did not lead to systematic bias in effect estimates. Because the study included a candidate gene methylation study of *NR3C1*, the chance of observing false positives and false negatives is high. The randomized experimental design minimizes the risk of spurious results in the study. To further minimize the risk of erroneous findings, future studies should consider using a combination of linkage mapping and a candidate gene approach.

In a low-resource setting, we found that an intensive combined nutrition, drinking water, sanitation, and handwashing intervention in early childhood reduced oxidative stress, enhanced the cortisol response, and reduced methylation levels of the glucocorticoid receptor gene. These findings support the future design and optimization of targeted nutritional and environmental therapeutic

approaches that leverage physiologic plasticity to improve children's health outcomes through the life course.

## Methods

### Study design and randomization

The WASH Benefits trial was conducted in rural villages in the Gazipur, Mymensingh, Tangail, and Kishoreganj districts of Bangladesh[28]. The traditional household structure in rural Bangladesh is the compound, where patrilineal families live together and share a common courtyard, and additionally sometimes other resources such as a pond, water source, and latrine. Eight pregnant women who lived near each other were grouped into a study cluster, the unit of randomization, to make it easy for a single community health promoter to walk to each compound. A one km buffer zone around each cluster was enforced in order to prevent spillover from nearby clusters. Eight geographically adjacent clusters formed a block. Using a random number generator, an investigator at UC Berkeley (B.F.A.) block randomized each of the eight geographically adjacent clusters to the double-sized control arm or to one of the six intervention arms in the parent trial: water; sanitation; handwashing; combined water, sanitation, and handwashing (WSH); nutrition, or combined nutrition, water, sanitation, and handwashing (N + WSH). This study only assessed physiological stress, oxidative stress, and DNA methylation in the control and the N + WSH arm of the trial.

Since each intervention delivered had visible physical components (lipid-based nutrient supplement sachets, chlorine tablets and storage vessels, potties, latrines, sani-scoop hoes, and handwashing stations), participants and outcome assessors were not masked. However, laboratory investigators were masked to group assignments and four researchers (A.L., A.N.M., S.T.T., and L.K.), following the pre-registered analysis plan, conducted independent masked statistical analyses. Analyses were replicated once. Results were only unmasked after replication of masked analyses.

### Sample size and power calculations

Because the sample size calculations were based on the original environmental enteric dysfunction study[49], we assumed that this sample size would be sufficient to assess the stress response and DNA methylation outcomes of this substudy. To estimate the minimum detectable effect of nutrition and WSH interventions on sAA, cortisol, oxidative stress, blood pressure, heart rate, and NR3C1 methylation in the trial, we assumed 135 clusters (average of 5 children per cluster) would be enrolled in this substudy and used the standard deviations in Table 4.

With a range of cluster-level intra-class correlations for repeated measures (0.01 to 0.20), the trial would have 90% power to detect the differences between each intervention arm and the control arm outlined in Table 5.

### Ethics

Study protocols were approved by human subjects committees at icddr,b (PR-11063 and PR-14108), the University of California, Berkeley (2011-09-3652 and 2014-07-6561) and Stanford University (25863 and 35583). icddr,b organized a data safety monitoring committee that oversaw the trial. The study protocol is available as a Supplementary Note. Participants provided written informed consent. Because compensation for research participation can be perceived as coercive in low-income settings, instead, families received health information (e.g., blood group testing results) as a token of appreciation.

### Participants

Pregnant women in their first or second trimesters and their children were enrolled in the study between 31 May 2012 and 7 July 2013[28]. The trial enrolled pregnant women in the first two trimesters

**Table 4 | Standard deviations of stress outcomes used in power calculations**

| Marker | Standard deviation |
| --- | --- |
| sAA (U/ml/min) | 1046.9[54] |
| Cortisol (µg/dl) | 0.80[55] |
| iPF(2α)-III (ng/mg creatinine) | 0.2[23] |
| 2,3-dinor-iPF(2α)-III (ng/mg creatinine) | 7.8[23] |
| iPF(2α)-VI (ng/mg creatinine) | 2.0[23] |
| 8,12-iso-iPF(2α)-VI (ng/mg creatinine) | 4.6[23] |
| NR3C1 methylation status (%) | 3.6[56] |
| Heart rate (beats per minute) | 20.7[57] |
| Mean arterial pressure (mmHg) | 12.0[58] |

to increase the number of available participants in the study area and to address the inaccuracies of gestational age estimation using self-reported last menstrual period dates. Households that had plans to move in the following year or did not own their own home were excluded in order to minimize loss to follow-up. Households that utilized a water source with high iron were excluded to optimize the effectiveness of the chlorine-based water treatment intervention. The selected study area had low levels of groundwater iron and arsenic, as determined by data from the Department of Public Health Engineering, the British Geological Survey, the Department for International Development National Hydro-chemical Survey, and a survey conducted before the study began[36]. Study staff also conducted surveys where respondents self-reported if there was iron taste in their drinking water or iron staining of their water storage vessels. If the respondent was uncertain about the iron content of their drinking water, study staff used Aquatabs and a digital Hach Pocket Colorimeter II to test the water's chlorine demand. Households with residual chlorine levels below 0.2 mg/L after 30 min were excluded.

### Procedures

The control group did not receive intervention-related household visits. The intervention group received a combination of interventions including a nutrition intervention, a drinking water intervention, a sanitation intervention, and a handwashing intervention; hereafter, this combined intervention group will be referred to as the N + WSH group. Details of the combined intervention in the parent trial were previously described[28]. Briefly, the nutrition component of the combined intervention consisted of the provision of lipid-based nutrient supplements (LNS; Nutriset, France) that included ≥100% of the recommended daily allowance of 12 vitamins and 9 minerals with 9.6 g of fat and 2.6 g of protein daily for children 6–24 months old and age-appropriate maternal and infant World Health Organization (WHO)/Food and Agriculture Organization (FAO) nutrition recommendations (pregnancy–24 months)[36]. The drinking water component of the combined intervention included chlorine tablets (Aquatabs; Medentech, Ireland) and safe storage vessels for drinking water. The sanitation component of the combined intervention included child potties, sani-scoop hoes to remove feces, and a double pit latrine for all households. The handwashing component of the combined intervention included handwashing stations with soapy water bottles and detergent soap placed near the latrine and kitchen. To promote behaviors such as treating water, using latrines, and handwashing, local community health promoters visited clusters at least once per week during the first 6 months, and subsequently, at least once every 2 weeks.

**Table 5 | Unadjusted and adjusted minimum detectable effect range for each stress outcome**

| Marker | Unadjusted minimum detectable effect range[a] | Adjusted minimum detectable effect range[b] |
|---|---|---|
| sAA (U/ml/min) | 266.41–350.48 | 333.22–438.38 |
| Cortisol (µg/dl) | 0.20–0.27 | 0.26–0.34 |
| iPF(2α)-III (ng/mg creatinine) | 0.05–0.07 | 0.06–0.08 |
| 2,3-dinor-iPF(2α)-III (ng/mg creatinine) | 1.99–2.61 | 2.48–3.27 |
| iPF(2α)-VI (ng/mg creatinine) | 0.51–0.67 | 0.64–0.84 |
| 8,12-iso-iPF(2α)-VI (ng/mg creatinine) | 1.17–1.54 | 1.46–1.93 |
| *NR3C1* methylation status (%) | 0.92–1.21 | 1.15–1.51 |
| Heart rate (beats per minute) | 5.27–6.93 | 6.59–8.67 |
| Mean arterial pressure (mmHg) | 3.05–4.02 | 3.82–5.03 |

[a]Unadjusted calculations using a two-sided alpha of 5%.
[b]Adjusted for multiple comparisons using a two-sided alpha of 0.56% (0.05 / 9 outcomes).

One year after intervention, urine samples for oxidative stress analysis were collected in Briggs Pediatric Sterile U-Bags and preserved with 0.1% thimerosal[49].

Two years after intervention, we collected saliva specimens. All study activities took place in the children's homes. In our stress response protocol for cortisol and salivary alpha-amylase measurements, the acute stressor was a venipuncture and caregiver physical separation from the child. Children refrained from ingesting caffeinated products and medicine at least one hour before the venipuncture stress protocol. One hour before the venipuncture, the study team obtained consent and interviewed the caregiver about their child's medical history. Thirty minutes before the venipuncture, the study team measured maternal and child blood pressure and heart rate. The child's mouth was rinsed with drinking water 15–20 min prior to the venipuncture. During the period leading up to the venipuncture, the children typically played or slept. Using SalivaBio Children's Swabs (Salimetrics), three saliva samples were collected during the stress response protocol (5–8 min before stressor onset, 5 min after stressor onset, and 20 min after stressor onset). The stressor was administered at a median time of 10:15 am (IQR: 9:20 am – 11:48 am). For each child, the stressor (the venipuncture and the physical separation of the child from the caregiver) lasted a median of 7.5 min (IQR: 7.5, 7.5). Cortisol was measured at two time points: pre-stressor and 20 min post-stressor. Salivary alpha-amylase was also measured at two time points: pre-stressor and 5 min post-stressor. Additional saliva samples for epigenetic analysis were collected in Oragene kits (OGR-575) and shipped at ambient temperature to EpigenDx (Hopkinton, MA) for DNA methylation analysis of the *NR3C1* gene.

At two years, the resting heart rate of participants was measured with a finger pulse oximeter (Nonin 9590 Onyx Vantage) in triplicate, and systolic and diastolic blood pressure were measured with a blood pressure monitor (Omron HBP-1300) in triplicate.

**Prespecified outcomes**
Analyses were intention-to-treat. We compared the N + WSH arm versus the control arm separately at one year after intervention (median age 14 months) and two years after intervention (median age 28 months). Outcomes included the concentrations of four isomers of F2-isoprostanes [iPF(2α)-III; 2,3-dinor-iPF(2α)-III; iPF(2α)-VI; 8,12-iso-iPF(2α)-VI] measured at one year after intervention. Pre-stressor and post-stressor concentrations of salivary alpha-amylase and salivary cortisol were measured at two years after intervention (median age

28 months). The overall methylation level of the glucocorticoid receptor (*NR3C1*) exon 1F promoter and the difference in percentage methylation at NGFI-A transcription factor binding site (CpG site 12) in DNA samples were measured at two years. Systolic and diastolic blood pressure and resting heart rate were measured at two years.

**Oxidative stress biomarker measurements**
F2-isoprostane isomers−iPF(2α)-III, 2,3-dinor-iPF(2α)-III, iPF(2α)-VI, and 8,12-iso-iPF(2α)-VI− were quantified by liquid chromatography-tandem mass spectrometry (LC-MS/MS) at Duke University as previously described and optimized for the present study[23,50]. Urine creatinine (CR) concentration was measured to determine sample volume used for F2-isoprostane analysis. A larger urine volume (300 µL) was used in case of low CR (CR < 0.6 mg/mL; highly diluted urine) to ensure assay sensitivity, a medium volume of urine (200 µL) was used when 0.6 mg/mL <CR < 1 mg/mL, whereas a lower volume (100 µL) was used when CR was high (CR > 1 mg/mL) to decrease the matrix suppression effect on F2-isoprostane signals. To the appropriate volume of urine sample, 20 µL of 1 M HCl, 20 µL of 100 ng/mL internal standard mix [iPF(2α)-III-d4, 8,12-iso-iPF(2α)-VI-d11, iPF(2α)-VI-d4], and 1 mL of methyl tert-butyl ether (MTBE) was added and vigorously mixed in FastPrep (Thermo) for 3 × 45 seconds at speed 4. After centrifugation, 800 µL of ether layer was evaporated (nitrogen stream), reconstituted in 50 µL methanol and 70 µL mobile phase A (see below) and 50 µL injected into Shimadzu 20 A series / Applied Biosystems API 4000 QTrap LC/MS/MS instrument. Two C18 columns (Agilent Eclipse Plus, 150 × 4.6 mm and 50 × 4.6 mm, 1.8 µm) in series were used with 0.1% acetic acid as mobile phase A and methanol as mobile phase B delivered as 40–75% B gradient elution over 26 minutes. The mass spectrometer was operated in negative mode with the following MS/MS transitions (m/z): 353/193 [iPF(2α)-III], 357/197 [iPF(2α)-III-d4], 325/237 [2,3-dinor-iPF(2α)-III], 353/115 [iPF(2α)-VI and 8,12-iso-iPF(2α)-VI], 364/115 [iPF(2α)-VI-d11], and 357/115 [8,12-iso-iPF(2α)-VI-d4]. Lower limits of quantification (LLOQ > 80% accuracy) were 0.063, 0.31, 0.63, and 0.63 mg/mL for iPF(2α)-III, 2,3-dinor-iPF(2α)-III, iPF(2α)-VI, and 8,12-iso-iPF(2α)-VI, respectively. The concentration of F2-isoprostanes was adjusted for urinary creatinine (CR) to account for urine diluteness. Creatinine (CR) was measured after 1/1000 dilution of urine by deionized water, centrifugation, and direct injection into the LC/MS/MS system. Agilent Eclipse Plus 50 × 4.6 mm, 1.8 µm column was used for separation. CR and CR-d3 (internal standard) were measured at m/z = 114/44 and m/z = 117/47, respectively.

**Physiological stress and methylation measurements**
Pre-stressor and post-stressor salivary alpha-amylase and cortisol were measured following ELISA kit protocols at icddr,b (Salimetrics, Carlsbad, CA). The initial cortisol sample was undiluted, and the initial dilution was 1:200 for salivary alpha-amylase. Out-of-range specimens were rerun at higher or lower dilutions. The coefficient of variation for salivary alpha-amylase and cortisol outcomes was <10%.

Saliva samples that were to be used for the analysis of DNA methylation were collected in Oragene kits (OGR-575) and analyzed by EpigenDx (Hopkinton, MA). EpigenDx performed salivary DNA extraction from Oragene samples, sample bisulfite treatment, PCR amplification, and pyrosequencing and determined percent methylation[51]. Methylation levels were assessed across the entire glucocorticoid receptor (*NR3C1*) exon 1F promoter (consisting of 39 assayed CpG sites)[52].

First, DNA was extracted from 200 µL saliva using DNAdvance (Beckman Coulter) with the Biomek FXP liquid handler (Beckman Coulter) at EpigenDx (Hopkinton, MA). The NanoDrop 2000 (Thermo Fisher Scientific) was used to quantify the extracted DNA by OD 260/280.

Next, EpigenDx carried out pyrosequencing of bisulfite-treated DNA. Briefly, 500 ng of extracted genomic DNA was bisulfite treated

**Table 6 | *NR3C1* pyrosequencing methylation assay target region**

| Assay ID | Assay Location | From ATG | From TSS (ENST00000231509) | GRCh38 (-) | # of CpG | Amplicon Size (bp) |
|---|---|---|---|---|---|---|
| ADS8063-FS2 | Promoter Exon 1F | −3533 to −3509 | −683 to −659 | Chr5:143404372-143404348 | 5 | 221 |
| ADS1343-FS2 | Promoter Exon 1F | −3470 to −3406 | −620 to −556 | Chr5:143404309-143404245 | 12 | 217 |
| ADS1343-FS3 | Promoter Exon 1F | −3389 to −3352 | −539 to −502 | Chr5:143404228-143404191 | 9 | 217 |
| ADS1342-FS | Promoter Exon 1F | −3341 to −3275 | −491 to −425 | Chr5:143404180-143404114 | 10 | 112 |
| ADS749-FS | Promoter Exon 1F | −3260 to −3204 | −410 to −354 | Chr5:143404099-143404043 | 7 | 100 |

using the EZ DNA Methylation kit (Zymo Research, Inc., CA). The kit protocol was followed for purification and elution of the bisulfite treated DNA (final elution volume of 46 μL). PCR amplification was achieved using 1 μL of bisulfite treated DNA and 0.2 μM of each primer. To purify the final PCR product using sepharose beads, one primer was biotin-labeled and purified by high performance liquid chromatography.

After being bound to Streptavidin Sepharose HP (GE Healthcare Life Sciences), the immobilized PCR products were purified, washed, denatured with a 0.2 μM NaOH solution, and rewashed using the Pyrosequencing Vacuum Prep Tool (Pyrosequencing, Qiagen), according to the manufacturer's instructions. The *NR3C1* pyrosequencing methylation assay target region is listed in Table 6.

Purified single stranded PCR products were annealed to 0.5 μM of sequencing primer. Following the manufacturer's protocol, 10 μL of the PCR products were pyrosequenced on the PSQ96 HS System (Pyrosequencing, Qiagen). QCpG software (Pyrosequencing, Qiagen) was used to analyze the methylation status of each locus (CpG site) individually as an artificial C/T SNP. To calculate the methylation level at each CpG site, the following formula was used: the percentage of methylated alleles divided by the sum of all methylated and unmethylated alleles. To obtain the mean methylation level, the methylation levels of all measured CpG sites within the targeted region of the gene were used. To ensure detection of incomplete bisulfite conversion of the DNA, each experiment used non-CpG cytosines as internal controls. Other controls in each PCR included unmethylated and methylated DNA. To test for bias, unmethylated control DNA was combined with in vitro methylated DNA at several ratios (0%, 5%, 10%, 25%, 50%, 75%, and 100%), the mixed products were bisulfite-modified and underwent PCR, followed by pyrosequencing analysis.

## Statistical analysis

The analysis protocol was pre-registered on Open Science Framework on 14 August 2019. The pre-registered analysis protocol and replication files for the substudy are available (https://osf.io/9573v/). For data management, we used STATA version 14.2. Analyses were conducted using R statistical software version 3.6.1. All biomarker distributions were right-skewed and thus log-transformed. Percentages of methylation were also skewed and therefore logit-transformed.

An individual was classified as a responder to the acute stressor if the difference between pre- and post-stressor cortisol concentrations was a positive change of at least two times the lower limit of sensitivity of the assay (0.014 μg/dL) and a change of at least two times the average coefficient of variation between duplicate tests of the same sample (20%). An individual was classified as a non-responder to the acute stressor if the difference between pre- and post-stressor cortisol concentrations was a negative change with the same thresholds outlined above. An individual was classified as no change if the difference between pre- and post-stressor cortisol concentrations was between these two thresholds, where the difference was not larger than the inherent error in the assay.

We used targeted maximum likelihood estimation with influence curve-based standard errors accounting for clustered observations from the trial's geographic block-randomized design[53].

The randomization of assignment to trial arm resulted in balance in the observed covariates across arms so the primary analysis was unadjusted. For each comparison between arms, we also conducted two secondary adjusted analyses: adjusting for child age and sex assigned at birth only and adjusting for child, age, sex, and covariates found to be significantly related to each outcome (likelihood ratio test $P < 0.2$). Time of sampling was included as a covariate for salivary alpha-amylase and cortisol analyses only. The full list of covariates is included in the footnotes of the tables.

We conducted a prespecified analysis estimating interactions between child sex and the intervention since biological differences, differential care practices, or other behavioral practices may influence the effect of the N + WSH interventions.

To determine whether missing specimen rates were random, we compared rates of missing specimens across arms and compared characteristics of participants with missing specimens and those with full sets. To account for imbalances in missing outcomes across arms and potential bias due to informative censoring, we repeated the adjusted analysis using inverse probability of censoring weighting, using covariates to predict missing outcomes[47,48].

The trial was registered at ClinicalTrials.gov (NCT01590095).

## Reporting summary

Further information on research design is available in the Nature Portfolio Reporting Summary linked to this article.

## Data availability

The prespecified, registered statistical analysis plan and deidentified individual participant data generated in this study have been deposited in Open Science Framework (https://osf.io/9573v/). Source data are provided with this paper. The raw DNA sequencing data discussed in this publication have been deposited in NCBI's Gene Expression Omnibus and are accessible through GEO Series accession number GSE261098. The raw liquid chromatography-tandem mass spectroscopy data have been deposited in the re3data repository and are accessible (https://doi.org/10.7924//r49311p2m). The consort checklist for the study is included in the Supplementary Information. Source data are provided with this paper.

## Code availability

The code and replication files for the study are publicly available on Open Science Framework (https://osf.io/9573v/) and GitHub (https://github.com/washb-eed-substudies/wash-stress).

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

## Acknowledgements

We greatly appreciate the families who participated in the study and the dedication of the icddr,b staff who delivered the interventions and collected the data and specimens. This study was funded by Global Development grant OPPGD759 from the Bill & Melinda Gates Foundation to the University of California, Berkeley [J.M.C.] and by the National Institute of Allergy and Infectious Diseases of the National Institutes of Health [grant number K01AI136885 to A.L.]. icddr,b is grateful to the Governments of Bangladesh, Canada, Sweden, and the United Kingdom for providing core/unrestricted support. National Institute of Health (NIH) / National Cancer Institute (NCI) Comprehensive Cancer Center Grant [grant number P30-CA014236-47 to I.S.] provided support for the Duke Cancer Institute PK/PD Core Laboratory. The funders approved the study design, but were not involved in data collection, analysis, interpretation, or any decisions related to publication. The corresponding author had full access to all study data and final responsibility for the decision to submit for publication.

## Author contributions

A.L. drafted the research protocol and manuscript with input from all listed co-authors; she coordinated input from the study team throughout the project. A.L., A.N.M., M.Z.R., C.P.S., L.C.H., B.F.A., M.R., L.U., A.E.H., I.S., P.K., J.M.C., S.P.L. and D.A.G. developed the interventions and guided interpretation of the results. M.Z.R., D.I., I.S., L.Y., A.M., M.R.K., S.S. and G.S. performed the laboratory analyses. A.L., S.A., C.P.S., L.C.H., B.F.A., A.E.H., S.L.F., S.A., M.S.H., P.M., A.K.S., M.R., L.U., C.D.H., P.K., J.M.C. and S.P.L oversaw study implementation and responded to threats to validity. A.L., A.N.M., S.T.T., D.I., I.S., C.P.S., L.C.H., L.K., L.Y., A.M., B.F.A., A.E.H., I.S., P.K., J.M.C., S.P.L. and D.A.G. developed the analytical approach, conducted statistical analyses, constructed tables and figures, and interpreted results. All authors have read, contributed to, and approved the final version of the manuscript.

## Competing interests

The content is solely the responsibility of the authors and does not necessarily represent the official views of the National Institutes of Health. In the interest of full disclosure, Douglas Granger is the founder and chief scientific and strategy advisor at Salimetrics LLC and SalivaBio LLC and these relationships are managed by the policies of the committees on conflict of interest at the Johns Hopkins University School of Medicine and the University of California at Irvine. Liying Yan is the president of EpigenDx, Inc. Ann Meyer is the Associate Director of Operations at EpigenDx, Inc. The remaining authors declare no competing interests.

## Ethics & Inclusion statement

The study complied with all relevant ethical regulations. The research included local researchers and stakeholders throughout the research process. The research is locally relevant as determined in collaboration with local partners at icddr,b. Roles and responsibilities as outlined in the author contributions section were agreed amongst collaborators ahead of the research. Capacity-building was included in all aspects of the research including intervention implementation, data collection, data analyses, laboratory analyses, and manuscript development. The parent trial included plans for local partners to lead and publish manuscripts and to disseminate study findings at conferences and other forums. Study protocols were approved by human subjects committees at the International Center for Diarrhoeal Disease Research, Bangladesh (icddr,b) (PR-11063 and PR-14108), the University of California, Berkeley (2011-09-3652 and 2014-07-6561) and Stanford University (25863 and 35583). icddr,b organized a data safety monitoring committee that oversaw the trial. All researchers on the project have access to all biological materials stored in Bangladesh and the United States. We have taken local and regional research relevant to our study into account in the citations.

## Additional information

**Peer review information** : *Nature Communications* thanks Prudence Atukunda, Brie Reid and the other, anonymous, reviewer(s) for their contribution to the peer review of this work. A peer review file is available.

Audrie Lin [1] ✉, Andrew N. Mertens [2], Md. Ziaur Rahman[3], Sophia T. Tan[4], Dora Il'yasova[5], Ivan Spasojevic [5,6],
Shahjahan Ali [3], Christine P. Stewart [7], Lia C. H. Fernald[2], Lisa Kim[2], Liying Yan[8], Ann Meyer[8], Md. Rabiul Karim[3],
Sunny Shahriar[3], Gabrielle Shuman[2], Benjamin F. Arnold [9], Alan E. Hubbard [2], Syeda L. Famida[3], Salma Akther[3],
Md. Saheen Hossen[3], Palash Mutsuddi[3], Abul K. Shoab[3], Idan Shalev[10], Mahbubur Rahman [3], Leanne Unicomb[3],
Christopher D. Heaney[11], Patricia Kariger[2], John M. Colford Jr.[2], Stephen P. Luby [4,14] & Douglas A. Granger[12,13,14]

[1]Department of Microbiology and Environmental Toxicology, University of California, Santa Cruz, Santa Cruz, CA, USA. [2]School of Public Health, University of
California, Berkeley, Berkeley, CA, USA. [3]Infectious Diseases Division, International Centre for Diarrhoeal Disease Research, Bangladesh, Dhaka, Bangladesh.
[4]Division of Infectious Diseases and Geographic Medicine, Stanford University, Stanford, CA, USA. [5]Department of Medicine, Duke University, Durham,
NC, USA. [6]PK/PD Core Laboratory, Duke Cancer Institute, Durham, NC, USA. [7]Institute for Global Nutrition, University of California Davis, Davis, CA, USA.
[8]EpigenDx Inc., Hopkinton, MA, USA. [9]Francis I. Proctor Foundation, University of California, San Francisco, CA, USA. [10]Department of Biobehavioral Health,
Pennsylvania State University, University Park, PA, USA. [11]Department of Environmental Health and Engineering, Johns Hopkins University, Baltimore, MD, USA.
[12]Institute for Interdisciplinary Salivary Bioscience Research, University of California, Irvine, Irvine, CA, USA. [13]Department of Pediatrics, Johns Hopkins
University School of Medicine, Baltimore, MD, USA. [14]These authors contributed equally: Stephen P. Luby, Douglas A. Granger. ✉e-mail: audrielin@ucsc.edu

