## [Peer Review File · Nature Communications]

Reviewers' Comments:

Reviewer #1:

Remarks to the Author:

Below are my comments:

1. It is unclear what is the unit of randomization. The authors seem to suggest 8 geographically adjacent clusters form a randomization unit. If that's the case, the authors need to rewrite the randomization section and abstract and anywhere else to clarify this. And how many blocks are there? It is unclear what "block randomized clusters" in line 239-240 means.
2. Is this study primary analysis of the trial? If yes, please describe briefly the power procedure of the design. Otherwise, it is unclear what is the goal of the estimation. Also, given the clustering/unclear unit of randomization in the study design, is there any account in the study design? That will significantly affect power.
3. The tables all use the word compound which was never mentioned in the writing. It reads confusing as the unit and size of the compound is unclear. And compounds sound like the study unit and its relationship with "clusters" in the paper is unclear.
4. The table presentation needs improvement. For instance, the acres of agricultural land owned all have two digits for mean and sd, but the year of education, and the distance to primary water source are all integers with no digits at all.
5. Why are there 6 treatment arms described in the design section if the results are only presented to have trt vs control?
6. In line 342-343, what are the clusters accounted in the analysis. The authors mentioned "account for any repeated measures in the clusters", which sounds confusing. The real matter here is the multilevel structure in the randomization as well as the potential geographical correlation across locations.
7. The paper has a strange structure with methods section came the last after the discussion and ends abruptly with a short discussion on missing data but no results are offered. Isn't any discussion needed for a research paper describing trial results like this (if not for a protocol paper)?

Reviewer #2:

Remarks to the Author:

This study describes the effects of an integrated nutritional water, sanitation, and handwashing intervention on child physiological stress response, oxidative stress, and DNA methylation in a large cohort in the first 2 years of life. The findings are rigorous and novel and will move both fields of developmental psychobiology and integrated nutrition interventions forward into a better understanding of developmental stress biology and how integrated interventions impact health and development. The writing is clear and concise, and the methods appear sound.

I have a few minor requests to aid in the interpretation and replication of the work moving forward.

The timeline of the stressor is a bit unclear to me. When were experimenters first interacting with the participating child (i.e., how long before the stressor onset were the children in the lab/interacting with experimenters)? There are elevations in cortisol levels when coming into a new environment, and an understanding of how long the child's cortisol levels were given to return to a pre-stressor baseline, along with an understanding of what the child was doing in the 20-30 minutes before the stressor would be helpful.

What time of day was the stressor administered, as diurnal cortisol patterns influence cortisol

levels in response to a stressor. Was time of sampling tested as a covariate?

How long was the stressor? E.g., how long were the children separated from their caregiver?

I was somewhat surprised to see that there was cortisol reactivity at the 1-year and 2-year mark—these are typically difficult ages to get a cortisol response from in response to caregiver separation (see Gunnar, 2009 for a review). I was also surprised that the effect sizes for cortisol were used in a newborn cohort (Keenan reference 49) rather than a toddler cohort. Studies that have seen elevated cortisol response in this later age group tend to be fearful, insecurely attached toddlers that exhibit elevations in cortisol to brief parental separations at an age when children don't elevate a cortisol response. Do the authors believe that the combined venipuncture and separation is the key? Were a proportion of children non-responders to the stressor? If possible, a plot of the cortisol trajectories by intervention group would be helpful.

- "Several of the studies did note increases in cortisol for insecurely attached children who were also highly fearful (Spangler and Gorssman, 1993; Gunnar et al., 1996b; Nachmias et al., 1996)." from Gunnar, 2009

- "Blood draws and inoculations become less reliable stressors by the second year of life. Indeed, only one of the seven studies of children between 13 months and 17 years reported mean increases in cortisol to blood draws or inoculations" (Gunnar, 2009)

- A recent inoculation study only found cortisol increases in response to the shot in insecurely attached toddlers below 150% of the federal poverty line (in the USA). (Johnson, 2017).

- Taken together, I think additional contextualization of what it means to have a higher cortisol response in this population, and what this might mean for the children moving forward would be worthwhile in the discussion. Especially as it relates to the higher starting cortisol levels for the female toddlers in the intervention arm - this is an interesting finding that was not addressed. In this paper, it seems that increased reactivity in response to the intervention is positive. While I don't disagree with this interpretation, some additional nuance in interpretation would be beneficial.

I would like to see some additional discussion on how the authors think the intervention is acting on child stress biology - is it through reductions in stress or increases in social support for the caregivers? The WASH Benefits trial did find lower levels of maternal depressive symptoms in all intervention groups compared to the control group in years 1 & 2 (Tofail, 2018), and this has been associated with blunted/flatter cortisol reactivity in infants (see Koss, 2017 for review). Or is it improved nutrition? I realize that you only analyzed the combined intervention arm, but my understanding is that length for age z-scores did not differ between children who received water, sanitation, handwashing, and nutrition and those who received the nutrition-only intervention (Luby 2018).

A minor point, but this is essentially a candidate gene study of NR3C1 methylation. As a proof of concept, I agree that this measure should be included in the paper and is important to consider. Nevertheless, it should be acknowledged as a limitation that these types of studies have been found to produce false positives

Reviewer #3:

None

Reviewer #4:

Remarks to the Author:

Alex McConnachie, Statistical Review

The paper by Lin et al reports the results of a substudy of a larger cluster randomised trial in rural Bangladesh, to assess the impact of an integrated nutritional, water, sanitation, and handwashing intervention on stress and epigenetic biomarkers. This review considers the statistical aspects of the paper.

The statistical methods used are good, and presented quite well, subject to some minor

observations.

In the abstract, the results for F2-isoprostanes are presented as a range of values, though for a reader unfamiliar with the biology (such as myself) it is not clear why. Could a few words be added to make it clear that there are several "F2-isoprostanes"?

The sample sizes achieved were quite a bit below the target, but this is not mentioned in the discussion. Should this at least be acknowledged?

In the abstract, the abbreviation "IPCW" is used without definition. I guessed what it meant, but others might not.

In the main paper, at line 343, it states that the (robust) standard errors accounted for repeated measures in the children. My understanding is that each child is only included in each analysis once. Does this statement actually refer to controlling for clustering of outcomes at the household or household-cluster level?

Line 350 refers to stratified analyses, though in reality I believe these are analyses with interaction terms to examine intervention effect differences between boys and girls.

The results are generally presented very clearly, though for me the third panel of Figure 2 does not work well, given the different measurement scales involved. Perhaps this needs a rethink.

Response to reviewer comments: NCOMMS-23-06611A, A randomized trial of water, sanitation, handwashing and nutritional interventions on stress and epigenetic programming

REVIEWER COMMENTS

Reviewer #1 (Remarks to the Author):

Below are my comments:

1. It is unclear what is the unit of randomization. The authors seem to suggest 8 geographically adjacent clusters form a randomization unit. If that's the case, the authors need to rewrite the randomization section and abstract and anywhere else to clarify this. And how many blocks are there? It is unclear what "block randomized clusters" in line 239-240 means.

Response: The unit of randomization was a study cluster. Eight geographically adjacent clusters were block-randomized to a double-sized control group or one of six different treatment groups. Eight pregnant women were grouped into a single cluster so that a single community health promoter could easily walk to the households. There was a total of 90 blocks of 8 clusters in the parent trial (720 clusters randomized), and 70 blocks in this substudy (135 clusters).

We have added and revised text in the Abstract, Methods – Study Design and Randomization section, and Results section to clarify this:

- Abstract revisions: *“Eight neighboring pregnant women were grouped into a study cluster. Eight geographically adjacent clusters were block-randomized into the control or the combined nutrition, water, sanitation, and handwashing (N+WSH) intervention group (receiving nutritional counseling and lipid-based nutrient supplements, chlorinated drinking water, upgraded sanitation, and handwashing with soap).”*
- Methods – Study Design and Randomization section revisions: *“The traditional household structure in rural Bangladesh is the compound, where patrilineal families live together and share a common courtyard, and additionally sometimes other resources such as a pond, water source, and latrine. Eight pregnant women who lived near each other were grouped into a study cluster, the unit of randomization, to make it easy for a single community health promoter to walk to each compound. A one km buffer zone around each cluster was enforced in order to prevent spillover from nearby clusters. Eight geographically adjacent clusters formed a block. Using a random number generator, an investigator at UC Berkeley (B.F.A.) block randomized each of the eight geographically adjacent clusters to the double-sized control arm or to one of the six intervention arms in the parent trial: water; sanitation; handwashing; combined water, sanitation, and handwashing (WSH); nutrition, or combined nutrition, water, sanitation, and handwashing (N+WSH).”*
- Results: *“A total of 90 blocks were defined, and each block was made up of 8 clusters randomly allocated to one of the intervention or control groups (720 clusters randomized) (Fig. 1).” “The substudy included 70 blocks (135 clusters).”*

2. Is this study primary analysis of the trial? If yes, please describe briefly the power procedure of the design. Otherwise, it is unclear what is the goal of the estimation. Also, given the clustering/unclear unit of randomization in the study design, is there any account in the study design? That will significantly affect power.

Response: This study was not the primary analysis of the trial. The parent trial was powered for diarrhea and linear growth primary outcomes. We used influence curve-based standard errors, treating individual study clusters as independent units, to account for the clustering-randomization in the trial, and have clarified that in the Methods:

“We used targeted maximum likelihood estimation with influence curve-based standard errors accounting for clustered observations from the trial’s geographic block-randomized design.”¹

¹ van der Laan M, Rose S. *Targeted Learning: Causal Inference for Observational and Experimental Data*. Springer Series in Statistics (2011).

The stress physiology outcomes were pre-specified and the sample size calculations were based on the original environmental enteric dysfunction substudy. ² Power calculations are detailed in the Supplementary Methods:

Because the sample size calculations were based on the original environmental enteric dysfunction study, ² we assumed that this sample size would be sufficient to assess the stress response and DNA methylation outcomes of this substudy. To estimate the minimum detectable effect of nutrition and WSH interventions on sAA, cortisol, oxidative stress, blood pressure, heart rate, and NR3C1 methylation in the trial, we assumed 135 clusters (average of 5 children per cluster) would be enrolled in this substudy and the following standard deviations:

Marker	Standard deviation
sAA (U/ml/min)	1046.9 ³
Cortisol (µg/dl)	0.80 ⁴
iPF(2α)-III (ng/mg creatinine)	0.2 ⁵
2,3-dinor-iPF(2α)-III (ng/mg creatinine)	7.8 ⁵
iPF(2α)-VI (ng/mg creatinine)	2.0 ⁵
8,12-iso-iPF(2α)-VI (ng/mg creatinine)	4.6 ⁵
NR3C1 methylation status (%)	3.6 ⁶
Heart rate (beats per minute)	20.7 ⁷
Mean arterial pressure (mmHg)	12.0 ⁸

With a range of cluster-level intra-class correlations for repeated measures (0.01 to 0.20), the trial would have 90% power to detect the following differences between each intervention arm and the control arm:

Marker	Unadjusted minimum detectable effect range*	Adjusted minimum detectable effect range**
sAA (U/ml/min)	266.41 – 350.48	333.22 – 438.38
Cortisol (µg/dl)	0.20 – 0.27	0.26 – 0.34
iPF(2α)-III (ng/mg creatinine)	0.05 – 0.07	0.06 – 0.08
2,3-dinor-iPF(2α)-III (ng/mg creatinine)	1.99 – 2.61	2.48 – 3.27
iPF(2α)-VI (ng/mg creatinine)	0.51 – 0.67	0.64 – 0.84
8,12-iso-iPF(2α)-VI (ng/mg creatinine)	1.17 – 1.54	1.46 – 1.93
NR3C1 methylation status (%)	0.92 – 1.21	1.15 – 1.51
Heart rate (beats per minute)	5.27 – 6.93	6.59 – 8.67
Mean arterial pressure (mmHg)	3.05 – 4.02	3.82 – 5.03

*Unadjusted calculations using a two-sided alpha of 5%

**Adjusted for multiple comparisons using a two-sided alpha of 0.56% (0.05 / 9 outcomes)

² Lin A, *et al.* Effects of Water, Sanitation, Handwashing, and Nutritional Interventions on Environmental Enteric Dysfunction in Young Children: A Cluster-randomized, Controlled Trial in Rural Bangladesh. *Clin Infect Dis* **70**, 738-747 (2020).

³ Kobayashi FY, *et al.* Salivary stress biomarkers and anxiety symptoms in children with and without temporomandibular disorders. *Braz Oral Res* **31**, e78 (2017).

⁴ Keenan K, Gunthorpe D, Young D. Patterns of cortisol reactivity in African-American neonates from low-income environments. *Dev Psychobiol* **41**, 265-276 (2002).

⁵ Il'yasova D, *et al.* Urinary biomarkers of oxidative status in a clinical model of oxidative assault. *Cancer Epidemiol Biomarkers Prev* **19**, 1506-1510 (2010).

⁶ Perroud N, *et al.* The Tutsi genocide and transgenerational transmission of maternal stress: epigenetics and biology of the HPA axis. *World J Biol Psychiatry* **15**, 334-345 (2014).

⁷ Garde A, *et al.* Respiratory rate and pulse oximetry derived information as predictors of hospital admission in young children in Bangladesh: a prospective observational study. *BMJ Open* **6**, e011094 (2016).

⁸ Ricci Z, Brogi J, De Filippis S, Caccavelli R, Morlacchi M, Romagnoli S. Arterial Pressure Monitoring in Pediatric Patients Undergoing Cardiac Surgery: An Observational Study Comparing Invasive and Non-invasive Measurements. *Pediatr Cardiol*, (2019).

3. The tables all use the word compound which was never mentioned in the writing. It reads confusing as the unit and size of the compound is unclear. And compounds sound like the study unit and its relationship with "clusters" in the paper is unclear.

Response: The traditional household structure in rural Bangladesh is the compound, where patrilineal families live together and share a common courtyard, and additionally sometimes other resources such as a pond, water source, and latrine. The number of people in a compound is variable. As mentioned above, the unit of randomization was a study cluster. Eight geographically adjacent clusters were block-randomized to a double-sized control group or one of six different treatment groups. Eight pregnant women were grouped into a single cluster so that a single community health promoter could easily walk to the households.

We have clarified and defined the relationship between compounds, clusters, and blocks in the Methods – Study Design and Randomization section: *“The traditional household structure in rural Bangladesh is the compound, where patrilineal families live together and share a common courtyard, and additionally sometimes other resources such as a pond, water source, and latrine. Eight pregnant women who lived near each other were grouped into a study cluster, the unit of randomization, to make it easy for a single community health promoter to walk to each compound. A one km buffer zone around each cluster was enforced in order to prevent spillover from nearby clusters. Eight geographically adjacent clusters formed a block. Using a random number generator, an investigator at UC Berkeley (B.F.A.) block randomized each of the eight geographically adjacent clusters to the double-sized control arm or to one of the six intervention arms in the parent trial: water; sanitation; handwashing; combined water, sanitation, and handwashing (WSH); nutrition, or combined nutrition, water, sanitation, and handwashing (N+WSH).”*

4. The table presentation needs improvement. For instance, the acres of agricultural land owned all have two digits for mean and sd, but the year of education, and the distance to primary water source are all integers with no digits at all.

Response: We have revised the table presentation so that the precision of the numeric values is consistent to one significant digit.

5. Why are there 6 treatment arms described in the design section if the results are only presented to have trt vs control?

Response: In this study, we only had funding to measure stress in the combined drinking water, sanitation, handwashing, and nutrition intervention arm versus the control arm. We included the description of the six treatment arms to provide the reader with the context of the parent trial.

We have revised the sentence in the Methods – Study Design and Randomization section to clarify this point: *“Using a random number generator, an investigator at UC Berkeley (B.F.A.) block randomized each of the eight geographically adjacent clusters to the double-sized control arm or to one of the six intervention arms in the parent trial: water; sanitation; handwashing; combined water, sanitation, and handwashing (WSH); nutrition, or combined nutrition, water, sanitation, and handwashing (N+WSH). This study only assessed physiological stress,*

oxidative stress, and DNA methylation in the control and the N+WSH arm of the trial.”

6. In line 342-343, what are the clusters accounted in the analysis. The authors mentioned "account for any repeated measures in the clusters", which sounds confusing. The real matter here is the multilevel structure in the randomization as well as the potential geographical correlation across locations.

Response: We have clarified that we used clustered standard errors to account for the cluster-randomized nature of the trial, controlling for potential dependence of outcomes at the geographic block level in the Methods. Although clusters are the unit of randomization, the geographic blocking could, in theory, induce some correlation in outcomes between clusters in the same block. Thus, we clustered the standard errors at the geographic block level, consistent with the primary analysis of the parent trial. We have revised the text in the manuscript as follows:

“We used targeted maximum likelihood estimation with influence curve-based standard errors accounting for clustered observations from the trial’s geographic block-randomized design.”¹

¹ van der Laan M, Rose S. *Targeted Learning: Causal Inference for Observational and Experimental Data*. Springer Series in Statistics (2011).

7. The paper has a strange structure with methods section came the last after the discussion and ends abruptly with a short discussion on missing data but no results are offer. Isn't any discussion needed for a research paper describing trial results like this (if not for a protocol paper)?

Response: The format of the sections in this article follows the typical format and ordering for *Nature Communications* articles: Abstract, Main, Results, Discussion, and Methods. This same format applies to trials, and a Discussion section was included in the original submission (please see lines 151-228 of the original submission).

We have included a discussion of the missing data results in the Results section of the manuscript: “*Unadjusted, adjusted, and inverse probability of censoring weighted (IPCW) analyses produced similar estimates (Supplementary Tables 2 and 3), indicating balance in measured confounders across arms and no differential loss to follow-up.*”

We have added a discussion of the missing data in the Discussion section of the manuscript: “*There was also loss-to-follow-up among children targeted for enrollment in this study and between measurement rounds (Fig. 1). Though loss-to-follow-up may have reduced the study’s power, the balance between household characteristics across intervention arms (both between children with outcome data and between those lost to follow-up; Supplementary Table 1), and the similarity of inverse probability of censoring weighting results to unadjusted and adjusted results, suggest that selection bias from differential loss-to-follow-up was unlikely.*”

Reviewer #2 (Remarks to the Author):

This study describes the effects of an integrated nutritional water, sanitation, and handwashing intervention on child physiological stress response, oxidative stress, and DNA methylation in a large cohort in the first 2 years of life. The findings are rigorous and novel and will move both fields of developmental psychobiology and integrated nutrition interventions forward into a better understanding of developmental stress biology and how integrated interventions impact health and development. The writing is clear and concise, and the methods appear sound.

I have a few minor requests to aid in the interpretation and replication of the work moving forward.

Response: Thank you for this helpful feedback.

The timeline of the stressor is a bit unclear to me. When were experimenters first interacting with the participating child (i.e., how long before the stressor onset were the children in the lab/interacting with experimenters)? There are elevations in cortisol levels when coming into a new environment, and an understanding of how long the child’s cortisol levels were

given to return to a pre-stressor baseline, along with an understanding of what the child was doing in the 20-30 minutes before the stressor would be helpful.

Response: Thank you for your suggestions to include these details to aid in replication and interpretation. The acute stressor was the child's physical separation from the caregiver and the venipuncture. Before the acute stressor was administered, all biological samples were collected at the child's home, so they were not introduced to a new environment. Approximately 1 hour before the acute stressor, the study team interacted with the caregiver to introduce themselves, obtain consent, and conduct a medical history survey. Approximately 30 minutes before the acute stressor, the study team measured maternal and child blood pressure and used a pulse oximeter to take measurements of heart rate. Approximately 15-20 minutes before the onset of the acute stressor, the child was given a cup of water to rinse out their mouth. During the 20 minutes leading up to the acute stressor, the child typically played or slept. The baseline pre-stressor Salimetrics saliva measurement was collected 5-8 minutes before the onset of the acute stressor. Because the same protocol was followed for each child, any systematic differences related to the study activities potentially affecting pre-stressor cortisol levels would likely affect both study groups equally.

We have added these details to the Methods – Procedures section: *“All study activities took place in the children's homes. In our stress response protocol for cortisol and salivary alpha-amylase measurements, the acute stressor was a venipuncture and caregiver physical separation from the child. Children refrained from ingesting caffeinated products and medicine at least one hour before the venipuncture stress protocol. One hour before the venipuncture, the study team obtained consent and interviewed the caregiver about their child's medical history. Thirty minutes before the venipuncture, the study team measured maternal and child blood pressure and heart rate. The child's mouth was rinsed with drinking water 15-20 minutes prior to the venipuncture. During the period leading up to the venipuncture, the children typically played or slept. Using SalivaBio Children's Swabs (Salimetrics), three saliva samples were collected during the stress response protocol (5-8 minutes before stressor onset, 5 minutes after stressor onset, and 20 minutes after stressor onset). Cortisol was measured at two time points: pre-stressor and 20 minutes post-stressor.”*

What time of day was the stressor administered, as diurnal cortisol patterns influence cortisol levels in response to a stressor. Was time of sampling tested as a covariate?

Response: For each child, the stressor was administered at a median time of 10:15 am (IQR: 9:20 am – 11:48 am). Time of sampling was pre-specified and tested as a covariate for salivary alpha-amylase and cortisol analyses only (<https://osf.io/9573v/>). We have added these details to the Methods section and the footnotes of Table 3 and Supplementary Table 3:

Methods: *“The stressor was administered at a median time of 10:15 am (IQR: 9:20 am – 11:48 am).” “Time of sampling was included as a covariate for salivary alpha-amylase and cortisol analyses only.”*

Table 3 and Supplementary Table 3 Footnotes: *“For salivary alpha-amylase and cortisol analyses only, time of sampling was included as a covariate.”*

How long was the stressor? E.g., how long were the children separated from their caregiver?

Response: The median time of the acute stressor (when the phlebotomists physically separated the child from their caregiver and administered the venipuncture) was a median of 7.5 minutes (IQR: 7.5, 7.5).

We have added these details to the Methods section of the manuscript: *“For each child, the stressor (the venipuncture and the physical separation of the child from the caregiver) lasted a median of 7.5 minutes (IQR: 7.5, 7.5).”*

I was somewhat surprised to see that there was cortisol reactivity at the 1-year and 2-year mark- these are typically difficult ages to get a cortisol response from in response to caregiver separation (see Gunnar, 2009 for a review). I was also surprised that the effect sizes for cortisol were used in a newborn cohort (Keenan reference 49) rather than a toddler cohort. Studies that have seen elevated cortisol response in this later age group tend to be fearful, insecurely attached toddlers that exhibit elevations in cortisol to brief parental separations at an age when children don't elevate a cortisol

response. Do the authors believe that the combined venipuncture and separation is the key? Were a proportion of children non-responders to the stressor? If possible, a plot of the cortisol trajectories by intervention group would be helpful.

Response: Because this study was not the primary analysis of the trial and the parent trial was powered for diarrhea and linear growth primary outcomes, the sample size was based on the original environmental enteric dysfunction substudy.² We assessed all children in the control and combined nutrition, water, sanitation, and handwashing intervention arms of the original environmental enteric dysfunction substudy. The cortisol outcome was a pre-specified outcome. Because the sample size was already pre-determined by the environmental enteric dysfunction substudy, the power calculations and effect sizes referenced, including the Keenan *et al.* reference, were auxiliary information detailed in the registered pre-analysis plan and the Supplementary Methods and not central for determining sample sizes for this substudy.

² Lin A, *et al.* Effects of Water, Sanitation, Handwashing, and Nutritional Interventions on Environmental Enteric Dysfunction in Young Children: A Cluster-randomized, Controlled Trial in Rural Bangladesh. *Clin Infect Dis* **70**, 738-747 (2020).

We believe that the combined venipuncture and physical separation of the child from the caregiver was the key to the stress response protocol for cortisol among this population. 69% of the children were responders to the acute stressor, 13% were non-responders, and 18% experienced no change.

We have added this information and text to the Methods section of the manuscript: “An individual was classified as a responder to the acute stressor if the difference between pre- and post-stressor cortisol concentrations was a positive change of at least two times the lower limit of sensitivity of the assay (0.014 µg/dL) and a change of at least two times the average coefficient of variation between duplicate tests of the same sample (20%). An individual was classified as a non-responder to the acute stressor if the difference between pre- and post-stressor cortisol concentrations was a negative change with the same thresholds outlined above. An individual was classified as no change if the difference between pre- and post-stressor cortisol concentrations was between these two thresholds, where the difference was not larger than the inherent error in the assay.”

We have also added the following text to the Results section of the manuscript: “In terms of cortisol reactivity, 69% of the children were classified as responders to the acute stressor (exhibited an increase in pre- to post-stressor cortisol levels), 13% of the children were classified as non-responders to the acute stressor (exhibited a decrease in pre- to post-stressor cortisol levels), and 18% experienced no change in cortisol levels (Supplementary Fig. 2).”

As requested by the Reviewer, we have added a plot of the cortisol trajectories by intervention group to the Supplementary Information: **Supplementary Figure 2. Pre- and post-stressor cortisol concentrations by the Control arm and Nutrition + WSH arm.**

- “Several of the studies did note increases in cortisol for insecurely attached children who were also highly fearful (Spangler and Gorssman, 1993; Gunnar et al., 1996b; Nachmias et al., 1996).” from Gunnar, 2009
- “Blood draws and inoculations become less reliable stressors by the second year of life. Indeed, only one of the seven studies of children between 13 months and 17 years reported mean increases in cortisol to blood draws or inoculations” (Gunnar, 2009)
- A recent inoculation study only found cortisol increases in response to the shot in insecurely attached toddlers below 150% of the federal poverty line (in the USA). (Johnson, 2017).
- Taken together, I think additional contextualization of what it means to have a higher cortisol response in this population, and what this might mean for the children moving forward would be worthwhile in the discussion. Especially as it relates to the higher starting cortisol levels for the female toddlers in the intervention arm - this is an interesting finding that was not addressed. In this paper, it seems that increased reactivity in response to the intervention is positive. While I don't disagree with this interpretation, some additional nuance in interpretation would be beneficial.

Response: Although the Reviewer cites several studies in the Gunnar et al. review, the participants in those studies do not have similar exposures or characteristics compared to the children in this substudy. To date, we are not aware of studies with participants who have characteristics or life histories comparable to the children in this substudy where cortisol reactivity to a stressor has been assessed. Generally, we can predict cortisol reactivity based on theoretical considerations, but given the unique environment in rural Bangladesh and the contextual sensitivity of the HPA axis to life experience, our general theoretical assumptions may or may not apply in this geographical context. After discussion with the co-authors, we would like to avoid speculation of the meaning of these novel results in relation to the prior literature because there are multiple interpretations that could all be valid: (1) Theory would suggest that HPA axis reactivity to a stressor is positive, adaptive, and appropriate when the challenge (in this case, venipuncture and physical separation from the caregiver) is novel or unfamiliar. We could speculate that this challenge task is novel or unfamiliar to the young children in this study in rural Bangladesh. Therefore, higher cortisol reactivity after a challenge task could be viewed as a positive response to stress. (2) Alternatively, HPA axis reactivity habituates to repeated exposure to stressors over time. We could speculate that the children enrolled in this study have life histories that exposed them to similar types of experiences as the challenge task, and thus, cortisol reactivity could be viewed as a negative response because the children's HPA axis response did not habituate. (3) Furthermore, given the study children's life histories, the challenge task may not be novel or unique, and therefore, most children would not have a cortisol response. (4) Another key issue linked to cortisol reactivity is the link to the regulation of glucose. Based on their diets, nutritional status, life histories, and geographical contexts, children in our study may have different glucose levels or issues regulating glucose or insulin, and we would anticipate observing differences in cortisol levels and reactivity. These different nutritional and metabolic contextual circumstances may complicate the direct comparison of these cortisol results to prior literature. Because the study of cortisol reactivity in this pediatric population in rural Bangladesh is quite novel and for the reasons outlined above, we hesitate to overinterpret these findings in the context of prior literature that has mostly taken place in high-income countries. Instead, to be conservative, we opt to limit the interpretation in the Discussion section to comparisons between the control and intervention arms.

Although child sex was an effect modifier and female children in the intervention group experienced elevated pre-stressor cortisol levels compared with female children in the control group, given the reasons outlined above, we hesitate to overinterpret these findings in the context of prior literature. Instead, in the Discussion section, we have added some text discussing this result and general implications: *“Sex assigned at birth was an effect modifier: among females, the intervention group experienced an elevation in pre-stressor cortisol levels. Identifying mediators of these sex-based differences in stress response will guide the future design of targeted early childhood interventions.”*

I would like to see some additional discussion on how the authors think the intervention is acting on child stress biology - is it through reductions in stress or increases in social support for the caregivers? The WASH Benefits trial did find lower levels of maternal depressive symptoms in all intervention groups compared to the control group in years 1 & 2 (Tofail, 2018), and this has been associated with blunted/flatter cortisol reactivity in infants (see Koss, 2017 for review). Or is it improved nutrition? I realize that you only analyzed the combined intervention arm, but my understanding is that length for age z-scores did not differ between children who received water, sanitation, handwashing, and nutrition and those who received the nutrition-only intervention (Luby 2018).

Response: Thank you for this helpful suggestion that greatly strengthens the manuscript. We have added an additional paragraph in the Discussion section addressing the Reviewer's points:

“In the parent trial, improved child linear growth was only observed in the arms receiving the nutrition intervention (the nutrition alone arm and the combined N+WSH arm),⁹ and child neurodevelopment was improved in all intervention arms.¹⁰ Although the combined N+WSH intervention may have directly impacted the physiological stress system through improved nutritional status and infection reduction, the intervention may also have had indirect impacts through reductions in caregiver depression and enhanced caregiver social support. In the WASH Benefits parent trial, community health promoters visited households in the intervention arm to promote intervention-related behaviors (e.g., treating water, using latrines), but they did not visit households in the control arm. The trial reported lower levels of depressive symptoms among women in the intervention arm compared with the control arm.¹⁰ Maternal depression is associated with blunted cortisol reactivity in infants, mediated by increased family conflict and less responsive parenting.¹¹ Hence, the intervention-associated reductions in maternal depressive symptoms and increased caregiver support may have led to the higher cortisol reactivity observed among children in the intervention arm compared with the children in the control arm. Forthcoming studies will aim to elucidate the complex interplay between nutrition, infection, psychosocial factors, and the physiological stress response.”

⁹ Luby SP, *et al.* Effects of water quality, sanitation, handwashing, and nutritional interventions on diarrhoea and child growth in rural Bangladesh: a cluster randomised controlled trial. *Lancet Glob Health* **6**, e302-e315 (2018).

¹⁰ Tofail F, *et al.* Effect of water quality, sanitation, hand washing, and nutritional interventions on child development in rural Bangladesh (WASH Benefits Bangladesh): a cluster-randomised controlled trial. *Lancet Child Adolesc Health* **2**, 255-268 (2018).

¹¹ Koss KJ, Gunnar MR. Annual Research Review: Early adversity, the hypothalamic-pituitary-adrenocortical axis, and child psychopathology. *J Child Psychol Psychiatry* **59**, 327-346 (2018).

A minor point, but this is essentially a candidate gene study of NR3C1 methylation. As a proof of concept, I agree that this measure should be included in the paper and is important to consider. Nevertheless, it should be acknowledged as a limitation that these types of studies have been found to produce false positives.

Response: Thank you for this suggestion. We have included the following language in the Limitations section of the Discussion: *“Because the study included a candidate gene methylation study of NR3C1, the chance of observing false positives and false negatives is high. The randomized experimental design minimizes the risk of spurious results in the study. To further minimize the risk of erroneous findings, future studies should consider using a combination of linkage mapping and a candidate gene approach.”*

Reviewer #3 (Remarks to the Author):

REVIEWER COMMENTS

First, I would like to commend the authors for undertaking this important work. While generally the manuscript is well-written and structured. I have some input that needs to be addressed. My comments are directed in the following key sections following the suggested Nature Communications peer review policy. Thank you for the opportunity to review this paper

Response: Thank you for these helpful comments.

Abstract

1. It will be helpful to give brief meaning of what they term as healthy trajectories in the introduction sentence.

Response: In the Abstract, we have revised the sentence and clarified that healthy trajectories refer to healthy child growth and development trajectories: *“A regulated stress response is essential for healthy growth and development trajectories.”*

2. Chronic stress, in the form of poverty, may cause irreversible harm if it occurs during the early years of life, a period of rapid growth and development. -Indicate in brackets what is the age range for this early years of life. - What form of poverty is this? difficulty in meeting basic needs such as housing or food - Any relationship between the acute stressor i.e physical separation of the child from the caregiver and chronic stress?

Response: We have revised this sentence in the Introduction section to address the Reviewer's questions about age range and poverty: *"Chronic stress, in the form of undernutrition, infection, and psychosocial adversity, may cause irreversible harm if it occurs during the early years of life (under age 2 years), a period of rapid growth and development."*

We have added a sentence in the Introduction section to address the Reviewer's question about the relationship between the acute stressor and chronic stress: *"Exposure to chronic stress may alter a child's cortisol response to an acute physical stressor, which could indicate HPA axis dysregulation."*

Results

3. Can IPCW term be defined on the first use in the abstract under results section?

Response: We have defined the abbreviation IPCW as inverse probability of censoring weighting. We have revised this sentence in the Results section, where this abbreviation first appears in the manuscript text: *"Unadjusted, adjusted, and inverse probability of censoring weighting (IPCW) analyses produced similar estimates (Supplementary Tables 2 and 3), indicating balance in measured confounders across arms and no differential loss to follow-up."*

Discussion

4. These two sentences are confusing; can they be consistent with the use of the terms in these sentences under the discussion section?

- In a setting with high levels of environmental contamination and food insecurity
- In settings with inadequate water, sanitation, and hygiene infrastructure, infections are acquired early and frequently in childhood

Response: We have replaced the first sentence with the following: *"In a setting with high levels of food insecurity and inadequate water, sanitation, and handwashing infrastructure, the trial found that a combined nutrition, water, sanitation, and handwashing intervention reduced oxidative stress, enhanced HPA axis functioning, and reduced methylation levels of the NGFI-A binding site in the NR3C1 exon 1F promoter in young children."*

5. Can the authors be consistent with their intervention content description? At one point they mention the intervention group receiving nutritional counseling and lipid-based nutrient supplements, chlorinated drinking water, upgraded sanitation, and handwashing with soap and another point they compress the intervention to be a combined drinking water, sanitation, handwashing, and nutritional intervention?

Response: We revised the manuscript text throughout to ensure consistency in the intervention description.

Specifically, we revised the abstract to ensure consistency with the intervention content description: *"Eight geographically adjacent clusters were block-randomized into the control or the combined nutrition, water, sanitation, and handwashing (N+WSH) intervention group (receiving nutritional counseling and lipid-based nutrient supplements, chlorinated drinking water, upgraded sanitation, and handwashing with soap)."*

We revised the Introduction/Main text section: *"Previously, the WASH Benefits trial reported that children receiving a combined nutrition, water, sanitation, and handwashing intervention experienced better growth and neurodevelopment compared to children in the control group."^{9,10}*

We revised the Results and Discussion sections throughout by replacing "intervention" with "combined N+WSH intervention".

We revised the Methods section to improve the clarity of the detailed description of the combined nutrition, water, sanitation, and handwashing intervention (please see the response to #13 below).

6. What does the term visible hardware mean in this study?

Response: Visible hardware refers to the physical components of the household interventions. The nutrition intervention included sachets of lipid-based nutrient supplements. The drinking water intervention consisted of chlorine tablets and safe storage vessels for drinking water. The sanitation intervention included child potties, sani-scoop hoes to remove feces, and a double pit latrine. The handwashing intervention consisted of handwashing stations with soapy water bottles and detergent soap.

We have replaced visible hardware with the phrase “physical components” and revised the text explanation in the Methods – Study Design and Randomization section: *“Since each intervention delivered had visible physical components (lipid-based nutrient supplement sachets, chlorine tablets and storage vessels, potties, latrines, sani-scoop hoes, and handwashing stations), participants and outcome assessors were not masked.”*

Ethics

7. What was the clearance board in Bangladesh? Indicate its name.

Response: The name of the human subjects clearance board in Bangladesh was the International Centre for Diarrhoeal Disease Research, Bangladesh (icddr,b). We have revised this sentence in the Ethics & Inclusion Statement section: *“Study protocols were approved by human subjects committees at the International Centre for Diarrhoeal Disease Research, Bangladesh (icddr,b) (PR-11063 and PR-14108), the University of California, Berkeley (2011-09-3652 and 2014-07-6561) and Stanford University (25863 and 35583).”*

Participants

8. Any justification for including mothers in both their first and second trimesters?

Response: The trial enrolled pregnant women in both their first and second trimesters because the number of pregnant women available to participate in the study was limited. There were also inaccuracies in gestational age using the self-reported date of the last menstruation.

We have added this justification to the Methods – Participants section: *“The trial enrolled pregnant women in the first two trimesters to increase the number of available participants in the study area and to address the inaccuracies of gestational age estimation using self-reported last menstrual period dates.”*

9. How did the authors identify Households that utilized a water source with high iron?

Response: The study area was selected using data from the Department of Public Health Engineering, the British Geological Survey, the Department for International Development National Hydrochemical Survey, and a survey conducted before the study began. The study team also conducted surveys to collect self-reported data to identify households with high iron water content. If compound residents reported no iron taste in their drinking water nor iron staining of their water storage vessels, then they were enrolled in the study. If the respondent was unsure about the iron content of their drinking water, the study team used Aquatabs and a digital Hach Pocket Colorimeter II to test the water’s chlorine demand. Households with residual chlorine levels below 0.2 mg/L after 30 minutes were excluded from the study.

We have added these details to the Methods – Participants section: *“The selected study area had low levels of groundwater iron and arsenic, as determined by data from the Department of Public Health Engineering, the British Geological Survey, the Department for International Development National Hydrochemical Survey, and a survey conducted before the study began.¹² Study staff also conducted surveys where respondents self-reported if there was iron taste in their drinking water or iron staining of their water storage vessels. If the respondent was uncertain about the iron content of their drinking water, study staff used Aquatabs and a digital Hach Pocket*

Colorimeter II to test the water's chlorine demand. Households with residual chlorine levels below 0.2 mg/L after 30 minutes were excluded."

¹² Stewart CP, et al. Effects of lipid-based nutrient supplements and infant and young child feeding counseling with or without improved water, sanitation, and hygiene (WASH) on anemia and micronutrient status: results from 2 cluster-randomized trials in Kenya and Bangladesh. *Am J Clin Nutr* **109**, 148-164 (2019).

Procedures

10. What is a passive control group with no study activities? Is it study activities or intervention activities? Can the authors use another term instead of passive control?

Response: A passive control group is a control group with no intervention-related household visits. The group that was previously referred to as the "passive control group" is now simply called the "control group".

We have clarified this point in the Methods – Procedures section: "*The control group did not receive intervention-related household visits.*"

11. Specify whose infant nutrition recommendations you mean here, WHO or another body?

Response: The study used the World Health Organization (WHO)/Food and Agriculture Organization (FAO) infant nutrition recommendations.

We have added these details to the Methods – Procedures section: "*Briefly, the nutrition component of the combined intervention consisted of the provision of lipid-based nutrient supplements (LNS; Nutriset, France) that included $\geq 100\%$ of the recommended daily allowance of 12 vitamins and 9 minerals with 9.6 g of fat and 2.6 g of protein daily for children 6–24 months old and age-appropriate maternal and infant World Health Organization (WHO)/Food and Agriculture Organization (FAO) nutrition recommendations (pregnancy–24 months).*"¹²"

12. Specify the brand of the chlorine tablets used in this study.

Response: The brand of chlorine tablets used in this study was Aquatabs (Medentech, Ireland).

We have added this additional detail to the Methods – Procedures section: "*The drinking water component of the combined intervention included chlorine tablets (Aquatabs; Medentech, Ireland) and safe storage vessels for drinking water.*"

13. Revise this long sentence, it makes it hard to follow through "The N+WSH intervention group received a combination of interventions: nutrition intervention [(lipid-based nutrient supplements that included $\geq 100\%$ of the recommended daily allowance of 12 vitamins and 9 minerals with 9.6 g of fat and 2.6 g of protein daily for children 6–24 months old and age-appropriate maternal and infant nutrition recommendations (pregnancy–24 months)], water (chlorine tablets and safe storage vessels for drinking water), sanitation (child potties, sani-scoop hoes to remove feces, and a double pit latrine for all households), and handwashing (handwashing stations, including soapy water bottles and detergent soap, near the latrine and kitchen)"

Response: We have revised this sentence as follows: "*The intervention group received a combination of interventions including a nutrition intervention, a drinking water intervention, a sanitation intervention, and a handwashing intervention; hereafter, this combined intervention group will be referred to as the N+WSH group. Details of the combined intervention in the parent trial were previously described.*"⁹ *Briefly, the nutrition component of the combined intervention consisted of the provision of lipid-based nutrient supplements (LNS; Nutriset, France) that included $\geq 100\%$ of the recommended daily allowance of 12 vitamins and 9 minerals with 9.6 g of fat and 2.6 g of protein daily for children 6–24 months old and age-appropriate maternal and infant World Health Organization (WHO)/Food and Agriculture Organization (FAO) nutrition recommendations (pregnancy–24 months).*"¹² *The drinking water component of the combined intervention included chlorine tablets (Aquatabs; Medentech, Ireland) and safe storage vessels for drinking water. The sanitation component of the combined intervention included child potties, sani-scoop hoes to remove feces, and a double pit latrine for all*

households. The handwashing component of the combined intervention included handwashing stations with soapy water bottles and detergent soap placed near the latrine and kitchen.”

14. Fig 1 why did the control have 180 clusters and intervention 90 clusters? Did this have any impact of the presented results? If not, how is this clearly attended to by the analyses?

Response: The control arm in the main trial was double-sized to increase the precision of the statistical tests used to compare the 6 intervention arms to the control arm in the parent factorial-designed trial. The study design and rationale paper for the parent trial gives the explanation:

“The control arm is double sized because it will be used in multiple hypothesis tests and, given available information, a 2:1 allocation ratio is close to the optimal allocation that minimises the variance for the six tests planned under our first hypothesis.¹³”

¹³ Arnold BF, *et al.* Cluster-randomised controlled trials of individual and combined water, sanitation, hygiene and nutritional interventions in rural Bangladesh and Kenya: the WASH Benefits study design and rationale. *BMJ Open* 3, e003476 (2013).

For the subsample used in this study, we studied the effect of one intervention arm (N+WSH) so we enrolled a subset of the control arm clusters so that the sample size was similar to the intervention arm. Clusters are the unit of randomization in the analysis, and we treated geographically blocked pairs of clusters in this substudy as the independent unit in the analysis, consistent with the analysis methods in the original parent trial (though limited to fewer clusters that were enrolled in this substudy).

15. Fig 1 any reasons why 72 withdrew from the control arm vs 18 in the control arm? How did this impact the study trial?

Response: We do not have the reasons for why participants withdrew, and while the withdrawals were larger in the control arm compared to the intervention arm, the proportion of participants loss-to-follow-up for any reason was balanced across arms, and the baseline covariate values of those lost to follow-up within the substudy were not differential by arm (Supplementary Table 1). Additionally, the baseline covariate values were not imbalanced between the full main trial population and the measured substudy population, those loss-to-follow-up between substudy sampling rounds were balanced across arms (Supplementary Table 1), and the sensitivity inverse-probability of censoring weighting were similar to the unadjusted and adjusted results. Therefore, we do not believe that the withdrawals biased the study results.

We have added this discussion of loss-to-follow-up and how we tested the impact on study results to the limitations section of the Discussion:

‘There was also loss-to-follow-up among children targeted for enrollment in this study and between measurement rounds (Fig. 1). Though loss-to-follow-up may have reduced the study’s power, the balance between household characteristics across intervention arms (both between children with outcome data and between those lost to follow-up; Supplementary Table 1), and the similarity of inverse probability of censoring weighting results to unadjusted and adjusted results, suggest that selection bias from differential loss-to-follow-up was unlikely.’

Reviewer #4 (Remarks to the Author):

Alex McConnachie, Statistical Review

The paper by Lin et al reports the results of a substudy of a larger cluster randomised trial in rural Bangladesh, to assess the impact of an integrated nutritional, water, sanitation, and handwashing intervention on stress and epigenetic biomarkers. This review considers the statistical aspects of the paper.

The statistical methods used are good, and presented quite well, subject to some minor observations.

Response: Thank you for this valuable statistical review.

In the abstract, the results for F2-isoprostanes are presented as a range of values, though for a reader unfamiliar with the biology (such as myself) it is not clear why. Could a few words be added to make it clear that there are several “F2-isoprostanes”?

Response: Thank you for this suggestion. We have revised the Abstract and clarified that there were four F2-isoprostanes isomers measured; hence, the range of differences between the control and N+WSH group: *“We measured four F2-isoprostanes isomers (iPF(2α)-III; 2,3-dinor-iPF(2α)-III; iPF(2α)-VI; 8,12-iso-iPF(2α)-VI), salivary alpha-amylase and cortisol, and methylation of the glucocorticoid receptor (NR3C1) exon 1F promoter including the NGFI-A binding site. Compared with control, the N+WSH group had lower concentrations of F2-isoprostanes isomers (differences ranging from -0.16 to -0.19 log ng/mg of creatinine, P<0.01), elevated post-stressor cortisol (0.24 log µg/dl; P<0.01), higher cortisol residualized gain scores (0.06 µg/dl; P=0.023), and decreased methylation of the NGFI-A binding site (-0.04; P=0.037).”*

The sample sizes achieved were quite a bit below the target, but this is not mentioned in the discussion. Should this at least be acknowledged?

Response: We have added to the Discussion section a limitation around the observed loss to follow-up, and how we tested that missing outcomes did not appear to be differential by treatment arm:

“There was also loss-to-follow-up among children targeted for enrollment in this study and between measurement rounds (Fig. 1). Though loss-to-follow-up may have reduced the study’s power, the balance between household characteristics across intervention arms (both between children with outcome data and between those lost to follow-up; Supplementary Table 1), and the similarity of inverse probability of censoring weighting results to unadjusted and adjusted results, suggest that selection bias from differential loss-to-follow-up was unlikely.”

In the abstract, the abbreviation “IPCW” is used without definition. I guessed what it meant, but others might not.

Response: The abbreviation IPCW appears for the first time in the Results section. We have added the definition for IPCW to that section: *“Unadjusted, adjusted, and inverse probability of censoring weighted (IPCW) analyses produced similar estimates (Supplementary Tables 2 and 3), indicating balance in measured confounders across arms and no differential loss to follow-up.”*

In the main paper, at line 343, it states that the (robust) standard errors accounted for repeated measures in the children. My understanding is that each child is only included in each analysis once. Does this statement actually refer to controlling for clustering of outcomes at the household or household-cluster level?

Response: We have updated that sentence because you are correct that we did not have repeated measures within the analyses. We used influence curve-based standard errors that were clustered at the geographic block level to account for the cluster-randomized nature of the trial, controlling for potential dependence of outcomes at the geographic block level. This was consistent with the primary analysis of the parent trial. We have clarified that in the Methods:

“We used targeted maximum likelihood estimation with influence curve-based standard errors accounting for clustered observations from the trial’s geographic block-randomized design.”¹

¹ van der Laan M, Rose S. *Targeted Learning: Causal Inference for Observational and Experimental Data*. Springer Series in Statistics (2011).

Line 350 refers to stratified analyses, though in reality I believe these are analyses with interaction terms to examine intervention effect differences between boys and girls.

Response: The Reviewer is correct; to be able to estimate a p-value for the strength of interaction, we fit models with interactions terms, and then we used the model to predict intervention effects by child sex, rather than

estimating the intervention effects stratified by child sex (Supplementary Tables 4-5). We have updated the sentence in the Methods to clarify this:

“We conducted a pre-specified analysis estimating interactions between child sex and the intervention since biological differences, differential care practices, or other behavioral practices may influence the effect of the N+WSH interventions.”

The results are generally presented very clearly, though for me the third panel of Figure 2 does not work well, given the different measurement scales involved. Perhaps this needs a rethink.

Response: Based on the Reviewer’s comments, we have decided to delete Figure 2 (“Unadjusted mean differences between the Control arm and the Nutrition + WSH arm for all stress outcomes”) given all of the estimates in Figure 2 are also reported in Tables 2-3, and visualizing the different measurement scales could confuse readers around the magnitude of intervention effects across outcomes with different measurement scales. We also deleted the original Supplementary Fig. 1, as this figure would have a similar issue with different measurement scales because it presented the “Adjusted mean differences between the Control arm and the Nutrition + WSH arm for all stress outcomes”. All the estimates in the original Supplementary Fig. 1 are also reported in Supplementary Tables 2-3.

References

1. van der Laan M, Rose S. *Targeted Learning: Causal Inference for Observational and Experimental Data*. Springer Series in Statistics (2011).
2. Lin A, *et al.* Effects of Water, Sanitation, Handwashing, and Nutritional Interventions on Environmental Enteric Dysfunction in Young Children: A Cluster-randomized, Controlled Trial in Rural Bangladesh. *Clin Infect Dis* **70**, 738-747 (2020).
3. Kobayashi FY, *et al.* Salivary stress biomarkers and anxiety symptoms in children with and without temporomandibular disorders. *Braz Oral Res* **31**, e78 (2017).
4. Keenan K, Gunthorpe D, Young D. Patterns of cortisol reactivity in African-American neonates from low-income environments. *Dev Psychobiol* **41**, 265-276 (2002).
5. Il'yasova D, *et al.* Urinary biomarkers of oxidative status in a clinical model of oxidative assault. *Cancer Epidemiol Biomarkers Prev* **19**, 1506-1510 (2010).
6. Perroud N, *et al.* The Tutsi genocide and transgenerational transmission of maternal stress: epigenetics and biology of the HPA axis. *World J Biol Psychiatry* **15**, 334-345 (2014).
7. Garde A, *et al.* Respiratory rate and pulse oximetry derived information as predictors of hospital admission in young children in Bangladesh: a prospective observational study. *BMJ Open* **6**, e011094 (2016).
8. Ricci Z, Brogi J, De Filippis S, Caccavelli R, Morlacchi M, Romagnoli S. Arterial Pressure Monitoring in Pediatric Patients Undergoing Cardiac Surgery: An Observational Study Comparing Invasive and Non-invasive Measurements. *Pediatr Cardiol*, (2019).
9. Luby SP, *et al.* Effects of water quality, sanitation, handwashing, and nutritional interventions on diarrhoea and child growth in rural Bangladesh: a cluster randomised controlled trial. *Lancet Glob Health* **6**, e302-e315 (2018).
10. Tofail F, *et al.* Effect of water quality, sanitation, hand washing, and nutritional interventions on child development in rural Bangladesh (WASH Benefits Bangladesh): a cluster-randomised controlled trial. *Lancet Child Adolesc Health* **2**, 255-268 (2018).

11. Koss KJ, Gunnar MR. Annual Research Review: Early adversity, the hypothalamic-pituitary-adrenocortical axis, and child psychopathology. *J Child Psychol Psychiatry* **59**, 327-346 (2018).
12. Stewart CP, *et al.* Effects of lipid-based nutrient supplements and infant and young child feeding counseling with or without improved water, sanitation, and hygiene (WASH) on anemia and micronutrient status: results from 2 cluster-randomized trials in Kenya and Bangladesh. *Am J Clin Nutr* **109**, 148-164 (2019).
13. Arnold BF, *et al.* Cluster-randomised controlled trials of individual and combined water, sanitation, hygiene and nutritional interventions in rural Bangladesh and Kenya: the WASH Benefits study design and rationale. *BMJ Open* **3**, e003476 (2013).

Reviewers' Comments:

Reviewer #1:

Remarks to the Author:

I have reviewed the rebuttal letter and revised manuscript. The authors have adequately revised the manuscript to clarify the unit of randomization, incorporating detailed explanations in the Abstract, Methods, and Results sections. This resolves the initial confusion regarding block-randomized clusters. The authors also clarified that this study was not the primary analysis of the trial. They have elaborated on the power procedure based on the original environmental enteric dysfunction substudy, which is a satisfactory response. The relationship between compounds and clusters is now better defined, resolving the confusion noted in the initial review. The authors have addressed the inconsistency in the presentation of numeric values in tables by standardizing the precision to one significant digit. The manuscript now also includes a clearer explanation of why only the combined nutrition, water, sanitation, and handwashing (N+WSH) intervention arm was studied despite mentioning six treatment arms. The authors have added new information and analyses regarding cortisol reactivity, including a plot of cortisol trajectories by intervention group. However, more nuanced discussion on interpreting increased cortisol reactivity and its implications would strengthen the manuscript. Also, the paper uses many abbreviations/acronyms, which makes the reading quite challenging even though I am reviewing it for the second time. I suggest the authors reduce the number of acronyms unless necessary. Additionally, the manuscript should consistently use either abbreviations or full terms for technical concepts after the first mention. For instance, "sAA" is defined multiple times in the manuscript. Please also check the others.

Reviewer #2:

Remarks to the Author:

I want to reiterate how important this work is and thank the authors and scientific team for this contribution to the literature. All of my points were addressed thoughtfully and well. I have no further comments or suggestions for the authors.

Reviewer #3:

Remarks to the Author:

Most comments have been addressed what is not addressed can be decided upon by the journal editor.

Nature Communications NComm 23-06611A

REVIEWER COMMENTS

First, I would like to commend the authors for attentively revising the manuscript and addressing most of the comments. While generally the manuscript is well revised. I have some specific comments that must be clearly addressed and approved by the journal editor. I have no more specific comments after authors address what I have indicated in here.

Abstract

1. It will be helpful to give brief meaning of what they term as healthy trajectories in the introduction sentence.

Accepted revision

2. Chronic stress, in the form of poverty, may cause irreversible harm if it occurs during the early years of life, a period of rapid growth and development.

-Indicate in brackets what is the age range for this early years of life.

- What form of poverty is this? difficulty in meeting basic needs such as housing or food
- Any relationship between the acute stressor i.e physical separation of the child from the caregiver and chronic stress?

Accepted revision

Results

3. Can IPCW term be defined on the first use in the abstract under results section?

Accepted revision

Discussion

4. These two sentences are confusing; can they be consistent with the use of the terms in these sentences under the discussion section?

- In a setting with high levels of environmental contamination and food insecurity
- In settings with inadequate water, sanitation, and hygiene infrastructure, infections are acquired early and frequently in childhood

Revised, however sentence is too long for readers to follow through. Make two sentences that are short to read and follow.

5. Can the authors be consistent with their intervention content description? At one point they mention the intervention group receiving nutritional counseling and lipid-based nutrient supplements, chlorinated drinking water, upgraded sanitation, and handwashing with soap and another point they compress the intervention to be a combined drinking water, sanitation, handwashing, and nutritional intervention?

Accepted revision

6. What does the term visible hardware mean in this study?

Accepted revision

Ethics

7. what was the clearance board in Bangladesh? Indicate its name.

Accepted revision

Participants

8. Any justification for including mothers in both their first and second trimesters? Accepted revision

9. How did the authors identify Households that utilized a water source with high iron?

Accepted revision

Procedures

0. What is a passive control group with no study activities? Is it study activities or intervention activities? Can the authors use another term instead of passive control?

Accepted revision

1. Specify whose infant nutrition recommendations you mean here, WHO or another body? Accepted revision

2. Specify the brand of the chlorine tablets used in this study.

Accepted revision

3. Revise this long sentence, it makes it hard to follow through “The N+WSH intervention group received a combination of interventions: nutrition intervention [(lipid-based nutrient supplements that included $\geq 100\%$ of the recommended daily allowance of 12 vitamins and 9 minerals with 9.6 g of fat and 2.6 g of protein daily for children 6–24 months old and age-appropriate maternal and infant nutrition recommendations (pregnancy–24 months)], water (chlorine tablets and safe storage vessels for drinking water), sanitation (child potties, sani-scoop hoes to remove feces, and a double pit latrine for all households), and handwashing (handwashing stations, including soapy water bottles and detergent soap, near the latrine and kitchen)”

Accepted revision

4. Fig 1 why did the control have 180 clusters and intervention 90 clusters? Did this have any impact of the presented results? If not, how is this clearly attended to by the analyses?

**“The subset of the control arm clusters so that the sample size was similar to the intervention arm”
How similar was it? Can authors briefly indicate the identified similarities?**

5. Fig 1 any reasons why 72 withdrew from the control arm vs 18 in the control arm? How did this impact the study trial?

Explanation given is not satisfactory, it needs clear explanation especially on why a such a huge withdrawal difference of 72 vs 18 cant not have an effect on the results? I believe this needs to be clearly accounted for. The journal statistician should follow this.

Reviewer #4:

Remarks to the Author:

Alex McConnachie, Statistical Review

I thank the authors for their consideration of my original comments. I am happy with their responses, and have no further comments to make.

Response to reviewer comments: NCOMMS-23-06611B, A cluster-randomized trial of water, sanitation, handwashing and nutritional interventions on stress and epigenetic programming

REVIEWERS' COMMENTS

Reviewer #1 (Remarks to the Author):

I have reviewed the rebuttal letter and revised manuscript. The authors have adequately revised the manuscript to clarify the unit of randomization, incorporating detailed explanations in the Abstract, Methods, and Results sections. This resolves the initial confusion regarding block-randomized clusters. The authors also clarified that this study was not the primary analysis of the trial. They have elaborated on the power procedure based on the original environmental enteric dysfunction substudy, which is a satisfactory response. The relationship between compounds and clusters is now better defined, resolving the confusion noted in the initial review. The authors have addressed the inconsistency in the presentation of numeric values in tables by standardizing the precision to one significant digit. The manuscript now also includes a clearer explanation of why only the combined nutrition, water, sanitation, and handwashing (N+WSH) intervention arm was studied despite mentioning six treatment arms. The authors have added new information and analyses regarding cortisol reactivity, including a plot of cortisol trajectories by intervention group. However, more nuanced discussion on interpreting increased cortisol reactivity and its implications would strengthen the manuscript. Also, the paper uses many abbreviations/acronyms, which makes the reading quite challenging even though I am reviewing it for the second time. I suggest the authors reduce the number of acronyms unless necessary. Additionally, the manuscript should consistently use either abbreviations or full terms for technical concepts after the first mention. For instance, “sAA” is defined multiple times in the manuscript. Please also check the others.

Response: Thank you for your thoughtful review and feedback. Because Reviewer #2 asked a similar question regarding cortisol reactivity and its implications, we are copying our response here to address your follow-up question:

The participants in existing studies in the stress literature do not have similar exposures or characteristics compared to the children in this substudy. To date, we are not aware of studies with participants who have characteristics or life histories comparable to the children in this substudy where cortisol reactivity to a stressor has been assessed. Generally, we can predict cortisol reactivity based on theoretical considerations, but given the unique environment in rural Bangladesh and the contextual sensitivity of the HPA axis to life experience, our general theoretical assumptions may or may not apply in this geographical context. After discussion with the co-authors, we would like to avoid speculation of the meaning of these novel results in relation to the prior literature because there are multiple interpretations that could all be valid: (1) Theory would suggest that HPA axis reactivity to a stressor is positive, adaptive, and appropriate when the challenge (in this case, venipuncture and physical separation from the caregiver) is novel or unfamiliar. We could speculate that this challenge task is novel or unfamiliar to the young children in this study in rural Bangladesh. Therefore, higher cortisol reactivity

after a challenge task could be viewed as a positive response to stress. (2) Alternatively, HPA axis reactivity habituates to repeated exposure to stressors over time. We could speculate that the children enrolled in this study have life histories that exposed them to similar types of experiences as the challenge task, and thus, cortisol reactivity could be viewed as a negative response because the children's HPA axis response did not habituate. (3) Furthermore, given the study children's life histories, the challenge task may not be novel or unique, and therefore, most children would not have a cortisol response. (4) Another key issue linked to cortisol reactivity is the link to the regulation of glucose. Based on their diets, nutritional status, life histories, and geographical contexts, children in our study may have different glucose levels or issues regulating glucose or insulin, and we would anticipate observing differences in cortisol levels and reactivity. These different nutritional and metabolic contextual circumstances may complicate the direct comparison of these cortisol results to prior literature. Because the study of cortisol reactivity in this pediatric population in rural Bangladesh is quite novel and for the reasons outlined above, we hesitate to overinterpret these findings in the context of prior literature that has mostly taken place in high-income countries. Instead, to be conservative, we opt to limit the interpretation in the Discussion section to comparisons between the control and intervention arms.

To improve clarity, we have reduced the number of abbreviations and deleted duplicate definitions for sAA.

Reviewer #2 (Remarks to the Author):

I want to reiterate how important this work is and thank the authors and scientific team for this contribution to the literature. All of my points were addressed thoughtfully and well. I have no further comments or suggestions for the authors.

Response: Thank you for your thoughtful review.

Reviewer #3 (Remarks to the Author):

Most comments have been addressed what is not addressed can be decided upon by the journal editor. See the attached document.

Response: Thank you for your thoughtful review and helpful feedback. We have addressed the remaining points here.

Discussion

*- In a setting with high levels of environmental contamination and food insecurity
- In settings with inadequate water, sanitation, and hygiene infrastructure, infections are acquired early
and frequently in childhood*

Revised, however sentence is too long for readers to follow through. Make two sentences that are short to read and follow.

Response: We have shortened the sentence by deleting the first part. The revised text reads as follows:

The trial found that a combined nutrition, water, sanitation, and handwashing intervention reduced oxidative stress, enhanced HPA axis functioning, and reduced methylation levels of the NGFI-A binding site in the NR3C1 exon 1F promoter in young children.

14. Fig 1 why did the control have 180 clusters and intervention 90 clusters? Did this have any impact of the presented results? If not, how is this clearly attended to by the analyses?

Original response: The control arm in the main trial was double-sized to increase the precision of the statistical tests used to compare the 6 intervention arms to the control arm in the parent factorial-designed trial. The study design and rationale paper for the parent trial gives the explanation:

“The control arm is double sized because it will be used in multiple hypothesis tests and, given available information, a 2:1 allocation ratio is close to the optimal allocation that minimises the variance for the six tests planned under our first hypothesis.¹”

¹ Arnold BF, *et al.* Cluster-randomised controlled trials of individual and combined water, sanitation, hygiene and nutritional interventions in rural Bangladesh and Kenya: the WASH Benefits study design and rationale. *BMJ Open* 3, e003476 (2013).

For the subsample used in this study, we studied the effect of one intervention arm (N+WSH) so we enrolled a subset of the control arm clusters so that the sample size was similar to the intervention arm. Clusters are the unit of randomization in the analysis, and we treated geographically blocked pairs of clusters in this substudy as the independent unit in the analysis, consistent with the analysis methods in the original parent trial (though limited to fewer clusters that were enrolled in this substudy).

“The subset of the control arm clusters so that the sample size was similar to the intervention arm” How similar was it? Can authors briefly indicate the identified similarities?

Additional response: The control and intervention arm subsamples were similar in the number of enrolled clusters (68 in the control arm, and 63 and 67 in the intervention arm in year 1 and 2, respectively), and in their baseline characteristics (Table 1, Supplementary Table 1). At enrollment, household characteristics were similar between the intervention and control arms (Table 1) and with the overall trial population (Supplementary Table 1). Therefore, even though this analysis utilizes a subset of the clusters enrolled in the full trial, the original randomization and exchangeability were maintained.

15. Fig 1 any reasons why 72 withdrew from the control arm vs 18 in the control arm? How did this impact the study trial?

Original response: We do not have the reasons for why participants withdrew, and while the withdrawals were larger in the control arm compared to the intervention arm, the proportion of participants loss-to-follow-up for any reason was balanced across arms, and the baseline covariate values of those lost to follow-up within the substudy were not differential by arm (Supplementary Table 1). Additionally, the baseline covariate values were not imbalanced between the full main trial population and the measured substudy population, those loss-to-follow-up between substudy sampling rounds were balanced across arms (Supplementary Table 1), and the sensitivity inverse-probability of censoring weighting were similar to the unadjusted and adjusted results. Therefore, we do not believe that the withdrawals biased the study results.

We have added this discussion of loss-to-follow-up and how we tested the impact on study results to the limitations section of the Discussion:

‘There was also loss-to-follow-up among children targeted for enrollment in this study and between measurement rounds (Fig. 1). Though loss-to-follow-up may have reduced the study’s power, the balance between household characteristics across intervention arms (both between children with outcome data and between those lost to follow-up; Supplementary Table 1), and the similarity of inverse probability of censoring weighting results to unadjusted and adjusted results, suggest that selection bias from differential loss-to-follow-up was unlikely.’

Explanation given is not satisfactory, it needs clear explanation especially on why a such a huge withdrawal difference of 72 vs 18 can’t not have an effect on the results? I believe this needs to be clearly accounted for. The journal statistician should follow this.

Additional response: We acknowledge that more children in the control group withdrew (n=72) compared with the intervention group (n=18), however, there are several results that we present in the paper that lead us to conclude the differential dropout did not lead to systematic bias in effect estimates. First, randomization was maintained (as evidenced by similar baseline characteristics between the intervention and control groups (Table 1)). Second, characteristics of children who were lost to follow-up at Year 2 were similar to those who remained (Supplementary Table 1). Finally, an inverse probability of censoring-weighted analysis, which re-weights the measured outcomes so the observed population reflects the characteristics of the full study population, led to very similar estimates (Supplementary Tables 2 and 3).^{2,3} Thus, the difference in withdrawal by arm was unlikely to bias our estimates, but did reduce the precision of our estimates by reducing the sample size.

We have revised the Discussion section of the manuscript to clarify these points and included new references (below):

Though loss-to-follow-up may have reduced the study's power, randomization was maintained, as evidenced by the balance between household characteristics across intervention and control arms (both between children with outcome data (Table 1) and between those lost to follow-up (Supplementary Table 1)), suggesting that selection bias from differential loss-to-follow-up was unlikely. Additionally, an inverse probability of censoring-weighted analysis, which re-weights the measured outcomes so the observed population reflects the characteristics of the full study population, produced similar estimates (Supplementary Tables 2 and 3), which further suggests that differential loss-to-follow-up likely did not lead to systematic bias in effect estimates.

² National Research Council, Division of Behavioral and Social Sciences and Education, Committee on National Statistics & Panel on Handling Missing Data in Clinical Trials. *The Prevention and Treatment of Missing Data in Clinical Trials*. (National Academies Press, 2010).

<https://www.ncbi.nlm.nih.gov/books/NBK209904/>

³ Little, R. J., D'Agostino, R., Cohen, M. L., Dickersin, K., Emerson, S. S., Farrar, J. T., Frangakis, C., Hogan, J. W., Molenberghs, G., Murphy, S. A., Neaton, J. D., Rotnitzky, A., Scharfstein, D., Shih, W. J., Siegel, J. P. & Stern, H. The prevention and treatment of missing data in clinical trials. *N. Engl. J. Med.* **367**, 1355–1360 (2012).

<https://pubmed.ncbi.nlm.nih.gov/23034025/>

Reviewer #4 (Remarks to the Author):

Alex McConnachie, Statistical Review

I thank the authors for their consideration of my original comments. I am happy with their responses, and have no further comments to make.

Response: Thank you for your thoughtful review.

References

1. Arnold BF, *et al.* Cluster-randomised controlled trials of individual and combined water, sanitation, hygiene and nutritional interventions in rural Bangladesh and Kenya: the WASH Benefits study design and rationale. *BMJ Open* 3, e003476 (2013).

2. National Research Council, Division of Behavioral and Social Sciences and Education, Committee on National Statistics & Panel on Handling Missing Data in Clinical Trials. *The Prevention and Treatment of Missing Data in Clinical Trials*. (National Academies Press, 2010).

<https://www.ncbi.nlm.nih.gov/books/NBK209904/>

3. Little, R. J., D'Agostino, R., Cohen, M. L., Dickersin, K., Emerson, S. S., Farrar, J. T., Frangakis, C., Hogan, J. W., Molenberghs, G., Murphy, S. A., Neaton, J. D., Rotnitzky, A.,

Scharfstein, D., Shih, W. J., Siegel, J. P. & Stern, H. The prevention and treatment of missing data in clinical trials. *N. Engl. J. Med.* **367**, 1355–1360 (2012).